# Pseudorabies virus inhibits progesterone-induced inactivation of TRPML1 to facilitate viral entry

**Bing-Qian Su**[1,2,3], **Guo-Yu Yang**[2,3,4], **Jiang Wang**[1,2,3,5]*, **Sheng-Li Ming**[1,2,3]*, **Bei-Bei Chu** [1,2,3,4,5,6]*

1 College of Veterinary Medicine, Henan Agricultural University, Zhengzhou, Henan Province, China, 2 Key Laboratory of Animal Biochemistry and Nutrition, Ministry of Agriculture and Rural Affairs, Zhengzhou, Henan Province, China, 3 Key Laboratory of Animal Growth and Development of Henan Province, Henan Agricultural University, Zhengzhou, Henan Province, China, 4 International Joint Research Center of National Animal Immunology, Henan Agricultural University, Zhengzhou, Henan Province, China, 5 Ministry of Education Key Laboratory for Animal Pathogens and Biosafety, Zhengzhou, Henan Province, China, 6 Longhu Advanced Immunization Laboratory, Zhengzhou, Henan Province, China

* wangjiang@henau.edu.cn (JW); mingsl911102@163.com (S-LM); chubeibei@henau.edu.cn (B-BC)

**Data Availability Statement:** All relevant data are within the manuscript and its Supporting Information files.

**Funding:** This study was supported by the National Key R&D Program of China: 2023YFD1801600 to

## Abstract

Viral infection is a significant risk factor for fertility issues. Here, we demonstrated that infection by neurotropic alphaherpesviruses, such as pseudorabies virus (PRV), could impair female fertility by disrupting the hypothalamus-pituitary-ovary axis (HPOA), reducing progesterone (P4) levels, and consequently lowering pregnancy rates. Our study revealed that PRV exploited the transient receptor potential mucolipin 1 (TRPML1) and its lipid activator, phosphatidylinositol 3,5-bisphosphate ($PI(3,5)P_2$), to facilitate viral entry through lysosomal cholesterol and $Ca^{2+}$. P4 antagonized this process by inducing lysosomal storage disorders and promoting the proteasomal degradation of TRPML1 via murine double minute 2 (MDM2)-mediated polyubiquitination. Overall, the study identifies a novel mechanism by which PRV hijacks the lysosomal pathway to evade P4-mediated antiviral defense and impair female fertility. This mechanism may be common among alphaherpesviruses and could contribute significantly to their impact on female reproductive health, providing new insights for the development of antiviral therapies.

## Author summary

Pseudorabies virus (PRV), an alphaherpesvirus of swine, is the causative agent of Aujeszky's disease and causes a significant economic impact in animal husbandry. Although it is known that PRV infection results in abortion, the mechanism involved in this clinical symptom remains elusive. Here, we reported that PRV infection affected fertility and progesterone (P4) levels through the hypothalamus-pituitary-ovary axis (HPOA). Moreover, PRV activated transient receptor potential mucolipin 1 (TRPML1) to facilitate lysosome-dependent viral entry. However, P4 induced proteasomal degradation of TRPML1 via murine double minute 2 (MDM2), thereby inhibiting viral entry. Overall, we have

B-BC and 2021YFD1301200 to G-YY. The funders
had no role in study design, data collection and
analysis, decision to publish, or preparation of the
manuscript.

**Competing interests:** The authors have declared
that no competing interests exist.

revealed a novel mechanism by which PRV influences P4 to induce infertility and promote viral replication.

## Introduction

Viral infections have been shown to be associated with reproductive health challenges, especially on female infertility. For example, embryos, particularly late blastocysts, appear to be susceptible to SARS-CoV-2 infection [1]. Progression of human immunodeficiency virus (HIV) disease results in a dramatic decline in both pregnancy and live birth rates [2–4]. Human papillomavirus can negatively impact embryo development by affecting the endometrial implantation of trophoblastic cells, leading to abnormal placentation and early pregnancy loss [5]. Additionally, herpes simplex virus (HSV) infection enhances HIV acquisition and transmission and has negative impacts on pregnancy rates, leading to increased incidences of miscarriage, abortion, and infertility [6]. Although vaccination is the optimal strategy to prevent viral infections, gaining a better understanding of how viruses induce infertility is crucial for managing public health crises.

The hypothalamus-pituitary-ovary axis (HPOA) is a complex network that regulates female fertility. Gonadotropin-releasing hormone (GnRH), secreted by the hypothalamus, is conveyed to the pituitary gland, which then releases follicle-stimulating hormone (FSH) and luteinizing hormone (LH) into the bloodstream. FSH and LH stimulate the growth and maturation of ovarian follicles. Sex steroids such as estrogen (E2) and progesterone (P4) are produced primarily by the ovaries [7]. P4 is critical for establishing and maintaining pregnancy by thickening the uterine lining [8]. Disorders of the HPOA can affect female fertility. Severe acute respiratory syndrome coronavirus 2 (SARS-CoV-2) infection induces a cytokine storm, which can affect the HPOA and may lead to pregnancy-related adverse events [9]. Nevertheless, the specific effects of viral infections on HPOA-related fertility have not been well documented.

Lysosomes function as cellular centers for signaling and metabolism [10]. It has been demonstrated that viruses can hijack lysosomes for optimal replication. We previously reported that the inhibition of Niemann-Pick type C1 (NPC1), a membrane protein involved in lysosomal cholesterol efflux [11], achieves broad antiviral activity. This occurs by decreasing cholesterol abundance on the plasma membrane (PM) and altering the dynamics of clathrin-coated pits (CCPs) [12]. However, membrane fusion mediated by filovirus glycoproteins and viral escape from the vesicular compartment require the NPC1 protein, independent of its known function in cholesterol transport [13]. β-coronaviruses use lysosomes for egress rather than the biosynthetic secretory pathway [14]. Lysosome inhibition by ammonium chloride and chloroquine disrupts HSV-1 multiplication [15], suggesting that lysosomes are necessary for HSV-1 replication. Nevertheless, the precise role of lysosomes in alphaherpesvirus infection remains a mystery.

We found that alphaherpesvirus acts on the HPOA to decrease P4 levels and consequent pregnancy rates. Mechanistic studies revealed that P4 induced the degradation of transient receptor potential mucolipin 1 (TRPML1, encoded by *MCOLN1*) to inhibit lysosome-dependent viral entry. The study highlights the intricate interplay between viruses and the lysosomal pathway, and suggests a potential target for antiviral therapies.

## Results

### PRV infection affects HPOA and fertility

Pseudorabies virus (PRV) is an ideal model for mechanistic investigations of alphaherpesviruses [16]. We first examined the effect of PRV infection on mouse mortality. Female

C57BL/6J mice were intranasally infected with PRV HN1201 (ranging from $1 \times 10^2$–$1 \times 10^4$ $TCID_{50}$ per mouse) for 12 days. No mice died when they were challenged with PRV at doses of $1 \times 10^2$ and $2 \times 10^2$ $TCID_{50}$ (S1A Fig). Therefore, in subsequent experiments, we intranasally infected the mice with PRV HN1201 at a dose of $2 \times 10^2$ $TCID_{50}$. To better delineate the histopathology induced by PRV infection in mice, we performed immunohistochemical staining for PRV glycoprotein E (gE) and hematoxylin and eosin staining of mouse tissues. PRV gE expression was detected in tissue sections from the ovary, uterus, hypothalamus, pituitary, and trigeminal nerve at two days post-infection with PRV (S1B Fig). Noticeable histopathological changes were observed in the ovary, uterus, pituitary, and trigeminal nerve, but not in the hypothalamus (S1B Fig). To determine whether PRV infection affected mouse fertility, mice were either mock-infected or infected with PRV, and then mated (Fig 1A). Subsequently, 9 out of 12 female mice in the control group successfully got pregnant 12 days post-mating. In contrast, in the PRV-infected group, 10 out of 12 did not become pregnant (Fig 1B). Additionally, both the number of embryos and the litter sizes were significantly lower in the PRV-infected group compared to those in the control group (Fig 1C and 1D). These results demonstrate that PRV infection adversely affects fertility in mice.

Given that PRV is a neurotropic alphaherpesvirus and the hypothalamic-pituitary-ovarian axis (HPOA) is a tightly regulated system controlling female fertility [17, 18], we hypothesized that PRV infection might impact the HPOA and thereby induce infertility. To test this hypothesis, we measured levels of HPOA-related hormones, including gonadotropin-releasing hormone (GnRH), luteinizing hormone (LH), follicle-stimulating hormone (FSH), progesterone (P4), and estradiol (E2), using enzyme-linked immunosorbent assay (ELISA) in the serum of mock- or PRV-infected mice. The ELISA results showed that levels of serum GnRH, LH, FSH, P4, and E2 were all increased in mock-infected mice during the 12-day pregnancy (Fig 1E–1I). Djati MS and colleagues have also reported that FSH levels in the serum of healthy pregnant mice rise steadily during the first 12 days of pregnancy [19]. In contrast, PRV infection resulted in significant decreases in the levels of GnRH, LH, FSH, P4, and E2 (Fig 1E–1I). Additionally, we evaluated the mRNA expression levels of the GnRH receptor (GnRHR) in the pituitary, and the LH/choriogonadotropin receptor (LHCGR) and FSH receptor (FSHR) in the ovary. Quantitative real-time polymerase chain reaction (qRT-PCR) analysis revealed that PRV infection downregulated the mRNA levels of GnRHR, LHCGR, and FSHR during pregnancy (S1C–S1E Fig). These data suggest that PRV infection interferes with normal HPOA function.

Leuprolide is a synthetic nonapeptide GnRHR agonist [20]. The effect of leuprolide on PRV-induced infertility was assessed (Fig 1J). ELISA results indicated that serum levels of LH, FSH, P4, and E2 in mock-infected mice were comparable to those in PRV-infected mice treated with leuprolide (S1F–S1I Fig). Consistently, the mRNA levels of the LHCGR and FSHR were also restored to control levels with leuprolide treatment (S1J and S1K Fig). When mice were infected with PRV, injection of leuprolide led to normal pregnancy rates, embryo numbers, and litter sizes (Fig 1K–1M). All results indicate that the viral infection affects the HPOA and induces infertility.

## PRV-induced infertility is caused by a decrease in P4 level

P4 is an important hormone that helps maintain a healthy pregnancy [21]. Given that PRV infection downregulated the pregnancy rate and P4 levels (Fig 1B and 1H), we determined whether P4 was pivotal in PRV-induced infertility (Fig 2A). No significant differences in body weight were observed in mice daily injected with P4 at dosages ranging from 0 to 10 mg/kg over 10 days (S2A Fig). Histological tissue analysis yielded similar results (S2B Fig). While

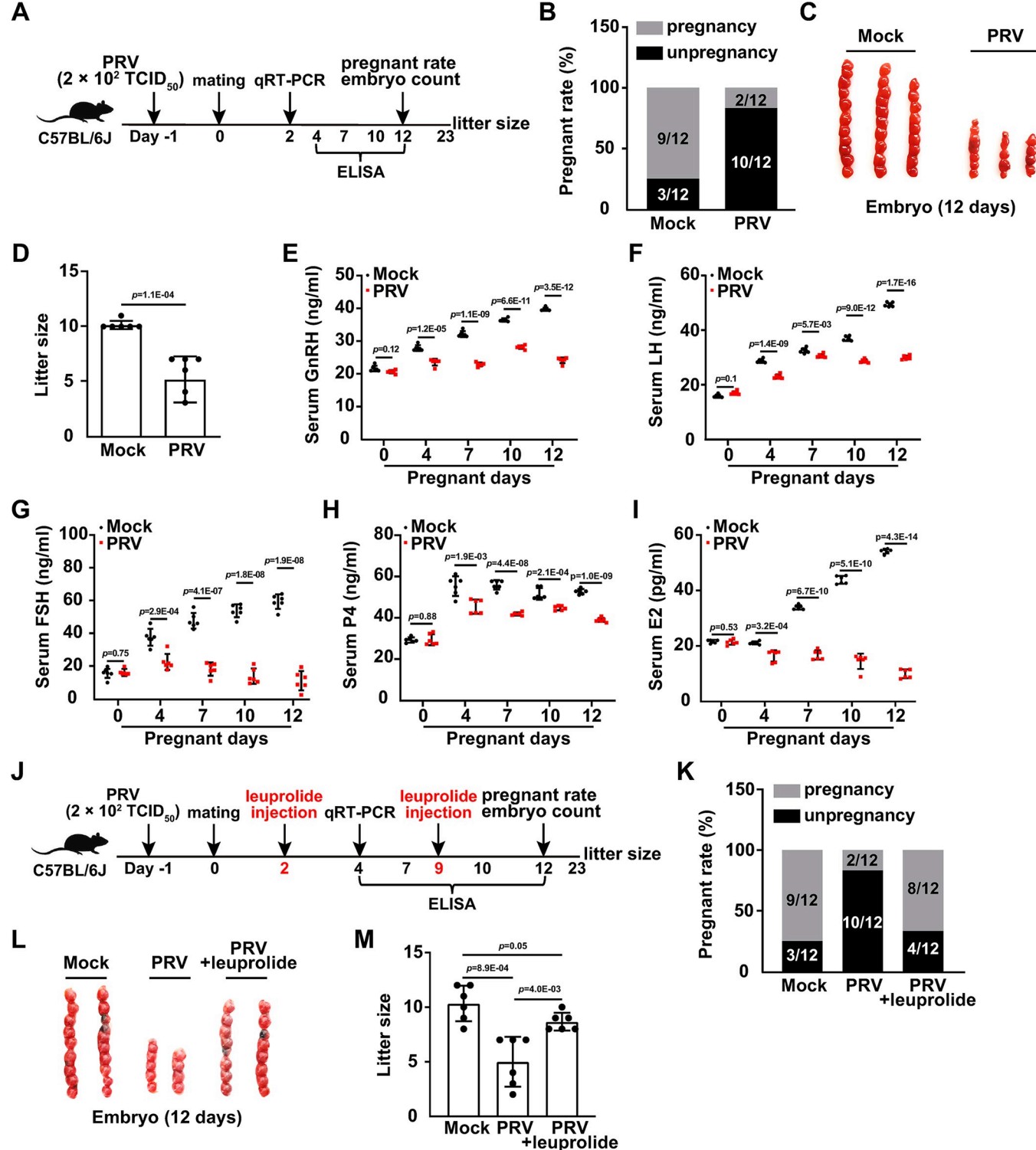

**Fig 1. PRV infection affects HPOA and fertility.** (A) Schematic diagram of the experimental procedure for PRV-induced infertility *in vivo*: On day -1, female C57BL/6J mice were either mock-infected or intranasally infected with PRV HN1201 ($2 \times 10^2$ $TCID_{50}$ per mouse). On day 0, both mock-infected and PRV-infected female mice were mated with male mice overnight. On day 2, the mRNA levels of GnRHR in the pituitary, and LHCGR and FSHR in the ovary, were analyzed by qRT-PCR. From day 4 to 12, GnRH, LH, FSH, P4, and E2 levels in serum were quantified by ELISA. The pregnancy rate and embryo numbers were determined on day 12. Litter size was counted on day 23. (B) The pregnancy rates of mock- or PRV-infected female mice were determined 12 days post-mating (n = 12). (C) The embryo number in mock- or PRV-infected female mice was determined 12 days post-pregnancy (n = 7). (D) The litter size of mock-

or PRV-infected female mice was determined after parturition (n = 7). (E–I) GnRH (E), LH (F), FSH (G), P4 (H) and E2 (I) levels in the serum of mock- or PRV-infected female mice were quantified by ELISA on the indicated days post-pregnancy (n = 6). (J) Schematic diagram of the experimental procedure for assessing the effect of leuprolide on PRV-induced infertility *in vivo*: On day -1, female C57BL/6J mice were either mock-infected or intranasally infected with PRV HN1201 ($2 \times 10^2$ TCID$_{50}$ per mouse). On day 0, both mock-infected and PRV-infected female mice were mated with male mice overnight. On day 2, female mice were injected with leuprolide (1.5 mg/kg). On day 4, the mRNA levels of GnRHR in the pituitary, as well as LHCGR and FSHR in the ovary, were analyzed by qRT-PCR. From day 4 to 12, serum levels of GnRH, LH, FSH, P4, and E2 were quantified by ELISA. Female mice were additionally injected with leuprolide (1.5 mg/kg per mouse) on day 9. Pregnancy rate and embryo numbers were determined on day 12. Litter size was calculated on day 23. (K) The pregnancy rate of the indicated female mice was determined 12 days post-mating (n = 12). (L) The embryo number in the indicated female mice was determined 12 days post-pregnancy (n = 7). (M) The litter size of the indicated female mice was determined after parturition (n = 7). Data are expressed as the mean ± SD of 3 independent experiments. *P*-values were determined by Student's *t*-test. $P < 0.05$ was considered statistically significant.

PRV infection downregulated serum P4 levels during pregnancy, injection with 10 mg/kg P4 restored P4 levels to the physiological norm (Fig 2B). P4 treatment dramatically alleviated PRV-induced infertility, as demonstrated by increased pregnancy rates, embryo numbers, and litter sizes (Fig 2C–2E). These results suggest that the infertility resulting from PRV infection is due to decreased P4 levels.

Considering that implantation is one of the most dramatic biological events during pregnancy [22], we investigated whether viral infection affected implantation. We observed fewer blue dye reactions in the uterus of PRV-infected mice compared to the control group, an effect that was reversed with the administration of exogenous P4 (Fig 2F). We also analyzed the mRNA levels of prostaglandin-endoperoxide synthase 2 (PTGS2) and secretin (SCT), which are involved in implantation, using qRT-PCR. PRV infection downregulated *Ptgs2* and *Sct* mRNA levels in the uterus, while P4 administration restored them to levels observed in the mock-infected group (Fig 2G and 2H). Similar results were observed in the uterus of leuprolide-challenged mice (S2C and S2D Fig). Collectively, these data demonstrate that PRV infection induces infertility by downregulating P4.

## P4 inhibits viral entry

Next, we assessed whether P4 influenced viral infection. Treatment of cells with P4 (0–10 μM) for 72 hours did not affect cell viability (S3A Fig). P4 is suggested to exert an immunosuppressive effect [23]. Therefore, we analyzed the effect of P4 on the expression of inflammatory cytokines in cells treated with immunostimulants (poly(I:C), poly(dA:dT), HT-DNA, and LPS). qRT-PCR analysis revealed that P4 prevented the immunostimulant-activated expression of IFN-β and IL-6 mRNA (S3B and S3C Fig). Conversely, P4 inhibited PRV infection *in vitro* (Fig 3A) and *in vivo*, as evidenced by reduced mortality associated with PRV infection (Fig 3B). These results suggest that P4's inhibitory effects on innate immunity are not implicated in attenuating viral infection. The effects of P4 are mediated by progesterone receptors (PGR), which regulate the expression of specific target genes [24]. Consequently, we determined whether P4 inhibits PRV infection through PGR. No significant difference in viral replication was observed between control and PGR knockout cells under P4 treatment (S3D and S3E Fig). P4 inhibited PRV gE expression in both control and PGR knockout cells, regardless of exogenous PGR expression (S3F Fig). These results indicate that P4 inhibits PRV infection in a receptor-independent manner.

To further investigate how P4 inhibited viral infection, cells were challenged with P4, and their viral attachment was analyzed. P4 treatment did not disrupt viral attachment, as indicated by qRT-PCR results measuring viral genome copy number and by Apollo staining of EdU-labeled PRV on the plasma membrane (PM) (S3G–S3I Fig). Additionally, viral entry was assessed using EdU-labeled PRV. P4 treatment inhibited viral entry, as shown in Fig 3C and 3D. The qRT-PCR analysis revealed that the incoming viral genome copy number was lower in P4-treated cells compared to control cells (Fig 3E). Transmission electron microscopy

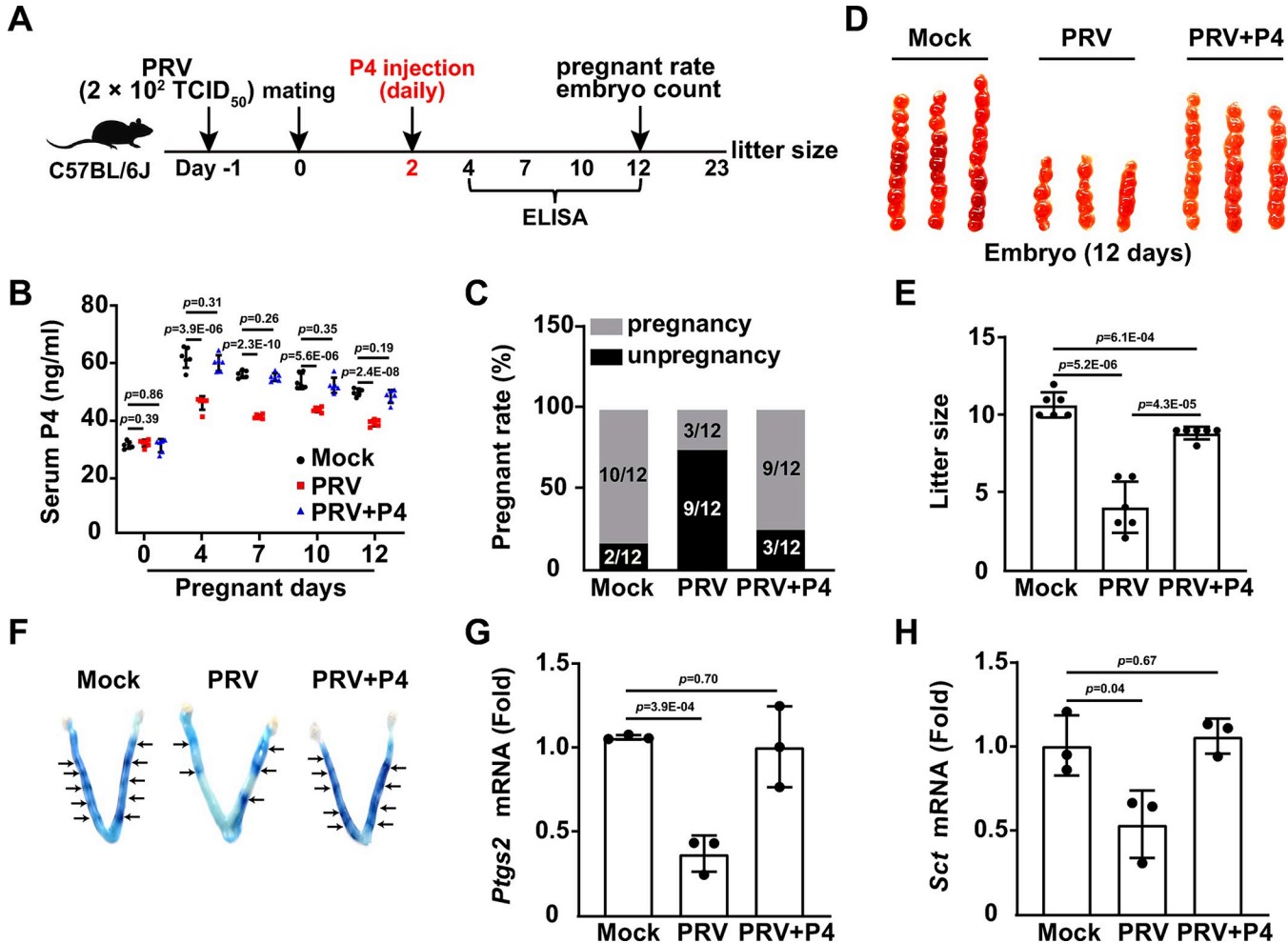

**Fig 2. PRV-induced infertility is caused by decreased P4 level.** (A) Schematic diagram of the experimental procedure for assessing the effect of P4 on PRV-induced infertility *in vivo*: On day -1, female C57BL/6J mice were either mock-infected or intranasally infected with PRV HN1201 ($2 \times 10^2$ TCID$_{50}$ per mouse). On day 0, both mock-infected and PRV-infected female mice were mated with male mice overnight. Starting on day 2, female mice were injected with P4 (10 mg/kg per mouse) daily for 10 days. From day 4 to 12, serum levels of GnRH, LH, FSH, P4, and E2 were quantified by ELISA. Pregnancy rates and embryo numbers were determined on day 12. Litter size was recorded on day 23. (B) P4 levels in serum of the indicated female mice were quantified by ELISA on the indicated days post-pregnancy (n = 6). (C) The pregnancy rate of the indicated female mice was determined 12 days post-mating (n = 12). (D) The number of embryos in the indicated female mice was determined 12 days post-pregnancy (n = 7). (E) The litter size of the indicated female mice was determined after parturition (n = 6). (F) Embryo distribution was observed on day 4.5 of pregnancy using the Chicago blue dye reaction. (G and H) PTGS2 and SCT mRNA levels in the uterus of the indicated female mice were analyzed using qRT-PCR. Data are expressed as the mean ± SD of 3 independent experiments. *P*-values were determined by Student's *t*-test. $P < 0.05$ was considered statistically significant.

results further demonstrated that P4 treatment resulted in virion retention at the PM (Fig 3F and 3G). Collectively, these results indicate that P4 inhibits viral entry.

As our previous study indicated that PRV entry is dependent on clathrin-coated pits (CCPs) [12], we analyzed whether P4 affected this endocytic pathway in the present study. Dynasore, a cell-permeable inhibitor of dynamin (DNM), prevents clathrin-mediated endocytosis [25]. As expected, dynasore treatment enhanced the association between the CCP adaptor-related protein complex 2 subunit beta 1 (AP2B1) and PRV gE on the PM, as indicated by the cell surface biotinylation assay (Fig 3H). Similarly, P4 treatment also resulted in enhanced interaction between AP2B1 and gE at the PM (Fig 3H). Fluorescence recovery after photobleaching (FRAP) analysis demonstrated that CCP dynamics, which are crucial for clathrin-

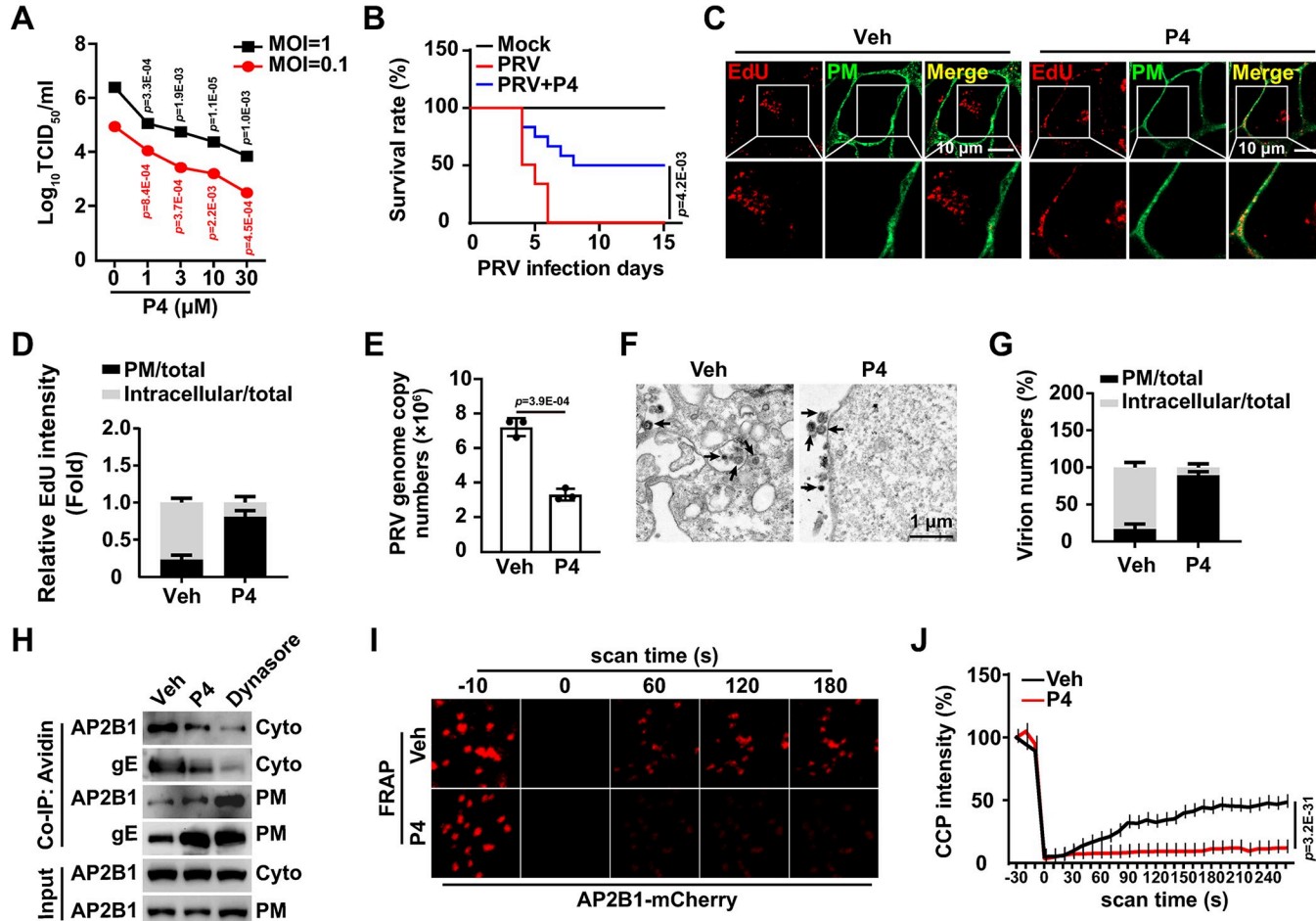

**Fig 3. P4 prevents viral entry.** (A) PK-15 cells were infected with PRV HN1201 (MOI = 0.1) and treated with P4 (0–30 μM) for 24 h. The viral titer was determined by the $TCID_{50}$ assay. (B) Mice were mock-infected, infected with PRV HN1201 ($2 \times 10^3$ $TCID_{50}$ per mouse), or infected with PRV HN1201 ($2 \times 10^3$ $TCID_{50}$ per mouse) and injected daily with P4 (10 mg/kg per mouse). The survival rate was monitored daily for 15 days (n = 12). (C) PK-15 cells were incubated with EdU-labeled PRV HN1201 (MOI = 0.1) and P4 (10 μM) at 4°C for 2 h. The cells were then shifted to 37°C for 10 min to allow entry. After washing with trypsin (1 mg/mL) to remove the residual virions on the PM, viral entry was detected by Apollo staining (red). DiI (green, 20 μM) indicated the PM. Scale bar: 10 μm. (D) Quantification of the relative total and intracellular EdU intensity in cells from (C) (n = 30). (E) PK-15 cells were incubated with PRV HN1201 (MOI = 0.1) and P4 (10 μM) at 4°C for 2 h. The cells were then shifted to 37°C for 10 min to allow entry. After washing with trypsin (1 mg/mL) to remove the residual virions on the PM, viral entry was detected by qRT-PCR analysis of viral genome copy numbers in the cells. (F) PK-15 cells were treated as in (E). Viral entry was detected by transmission electron microscopy. Scale bar: 1 μm. (G) Quantification of the total and intracellular virion numbers in PK-15 cells from (F) (n = 20). (H) PK-15 cells were treated with vehicle, P4 (10 μM), or dynasore (20 μM) for 24 h. Interactions of AP2B1 with PRV gE on the PM and in the cytosol were detected using a cell surface biotinylation assay. (I) PK-15 cells were transfected with an AP2B1-mCherry plasmid for 24 h and treated with vehicle or P4 (10 μM) for 8 h. CCP dynamics were detected using time-lapse confocal microscopy. (J) Quantification of the relative fluorescence intensity of AP2B1 puncta in the FRAP region over time from (I) (n = 10). Data are expressed as the mean ± SD of 3 independent experiments. *P*-values were determined by Student's *t*-test. *P* < 0.05 was considered statistically significant.

mediated endocytosis, were inhibited by P4 treatment (Fig 3I and 3J). Altogether, these data suggest that P4 inhibits viral entry by interfering with CCP function.

## P4 results in lysosomal storage disorders (LSDs)

We previously demonstrated that cholesterol is indispensable for CCP dynamics [12]. Therefore, we analyzed the cellular cholesterol level using filipin staining in cells treated with P4. Surprisingly, P4 treatment led to cholesterol accumulation in the lysosomes, but not in the mitochondria, peroxisomes, or endoplasmic reticulum (ER) (S4A Fig). Furthermore, lysosomal pH increased from approximately 4 to 7 due to P4 treatment, suggesting that P4 induced

lysosomal dysfunction (S4B Fig). Cholesterol export from lysosomes requires Niemann-Pick type C1 (NPC1), an integral membrane protein on the lysosomal membrane [11]. Overexpression of NPC1 reversed the P4-induced accumulation of cholesterol in the lysosomes (Fig 4A). Consequently, we examined the effect of NPC1 overexpression on viral entry. Although NPC1 expression mitigated the viral entry inhibition caused by P4, it failed to restore it fully to the levels observed in control cells, as indicated by qRT-PCR analysis of the PRV genome copy number in the cells (Fig 4B). Similar results were observed in cells supplemented with exogenous cholesterol (Fig 4C). These findings suggest that an additional mechanism may be involved in the inhibition of viral entry by P4.

Although cholesterol was exported from lysosomes upon NPC1 overexpression during P4 treatment, perinuclear redistribution of lysosomes was observed (S4C and S4D Fig), suggesting that the lysosomes were still abnormal. Given that lysosomes are a major intracellular $Ca^{2+}$ storage organelle increasingly implicated in regulating endocytosis [26], we hypothesized that P4 treatment led to $Ca^{2+}$ accumulation in lysosomes. Staining for intracellular $Ca^{2+}$ with Fluo-3 showed distribution throughout the cells (Fig 4D). P4 treatment induced perinuclear $Ca^{2+}$ accumulation, which co-localized with lysosomal-associated membrane protein 1 (LAMP1) (Figs 4D, S4E and S4F). When $Ca^{2+}$ was released from lysosomes using Gly-Phe β-naphthylamide (GPN), an increase in $Ca^{2+}$ release was observed in P4-treated cells compared to control cells (Fig 4E). NPC1 overexpression did not prevent the accumulation of $Ca^{2+}$ in lysosomes in P4-treated cells (Fig 4F). These findings suggest that P4 leads to $Ca^{2+}$ accumulation in lysosomes.

Next, we determined whether lysosomal $Ca^{2+}$ was involved in viral entry. Fluo-3 staining showed that PRV infection induced $Ca^{2+}$ transport from the cytoplasm to the PM, which was impeded by P4 (Fig 4G and 4H). PRV infection stimulated significant $Ca^{2+}$ increases, while $Ca^{2+}$ levels in P4-treated cells were comparable to control levels (Fig 4I). When $Ca^{2+}$ was released from lysosomes with GPN or Bafilomycin A1 (Baf A1), PRV did not induce an increase in $Ca^{2+}$ levels (Fig 4J). However, the release of $Ca^{2+}$ from the endoplasmic reticulum (ER) using cyclopiazonic acid (CPA) or from mitochondria using carbonyl cyanide 3-chlorophenylhydrazone (CCCP) did not affect PRV-induced $Ca^{2+}$ increases (Fig 4J). qRT-PCR analysis of the viral genome indicated that both P4 and the intracellular $Ca^{2+}$ chelator BAPTA AM inhibited viral entry (Fig 4K). These data suggest that PRV infection promotes lysosomal $Ca^{2+}$ transport to the PM to facilitate viral entry.

DNM mediates vesicle fission from the PM during endocytosis, while its dephosphorylation activates the intrinsic GTPase activity, which depends on $Ca^{2+}$ [27]. For this reason, we analyzed DNM1 phosphorylation by immunoblotting during viral entry. DNM1 became dephosphorylated as early as 5 min post-viral entry (Fig 4L). No change in DNM1 phosphorylation was observed in PRV-infected cells treated with P4 for 0–20 min (Fig 4L). These results indicate that P4 inhibits DNM dephosphorylation to prevent viral entry. To further confirm the involvement of cholesterol and $Ca^{2+}$ in viral entry, P4-treated cells were supplemented with cholesterol, $CaCl_2$, or both cholesterol and $CaCl_2$. Supplementation with either cholesterol or $CaCl_2$ partially rescued viral entry, whereas simultaneous treatment of cells with both cholesterol and $CaCl_2$ restored viral entry to control levels (Fig 4M). Viral titer assays indicated that cholesterol and $CaCl_2$ counteracted the inhibitory effect of P4 on viral replication (Fig 4N). Taken together, these results demonstrate that P4 causes LSDs to prevent viral entry.

## PRV infection upregulates transient receptor potential mucolipin 1 (TRPML1) expression

Lysosomal $Ca^{2+}$ release requires lysosomal $Ca^{2+}$ efflux channels, such as TRPML1 and two-pore channel (TPC) proteins [28]. We hypothesized that PRV infection might affect lysosomal

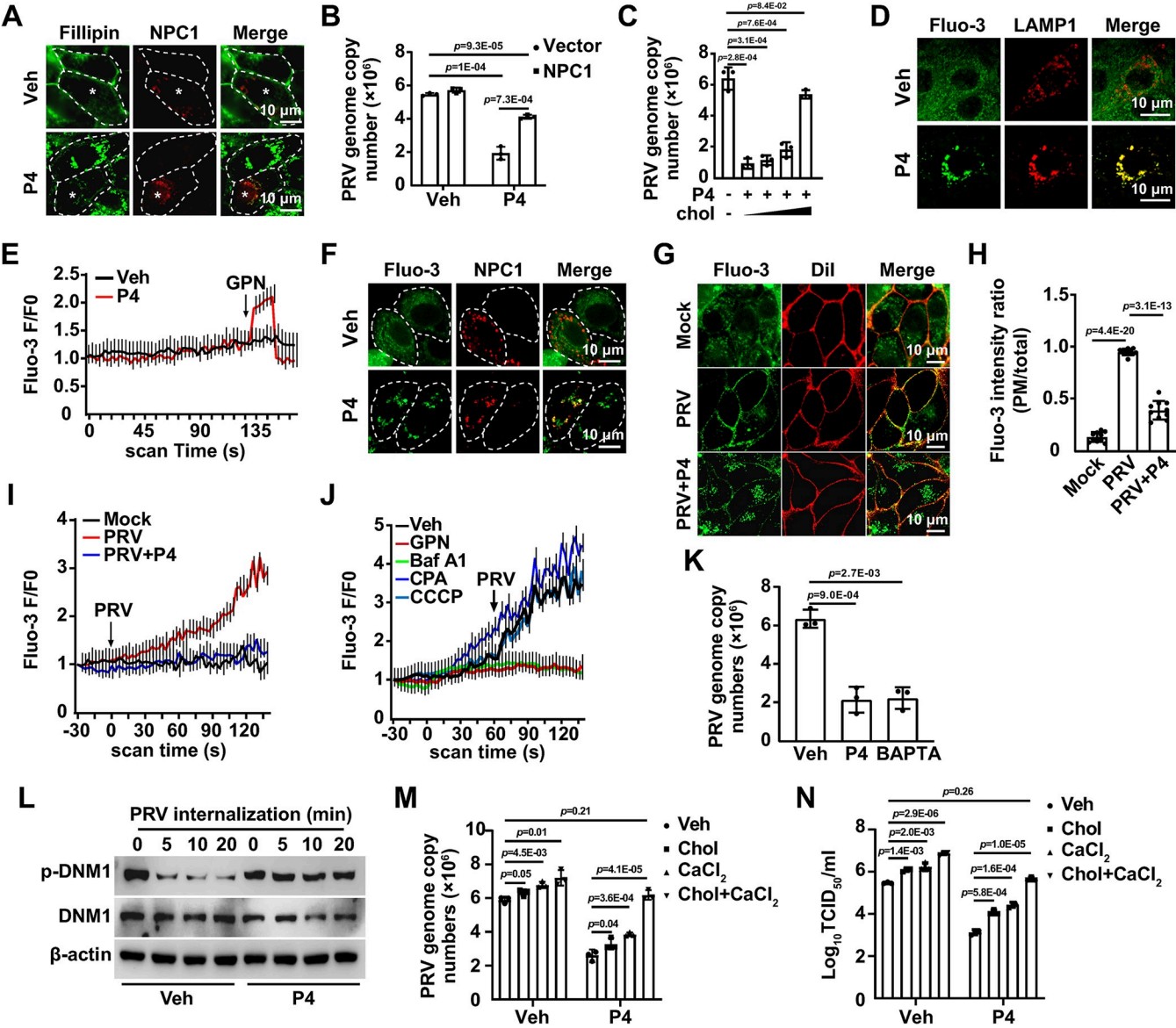

**Fig 4. P4 results in LSDs.** (A) PK-15 cells were transfected with NPC1-mCherry plasmid (red) for 24 hours followed by treatment with vehicle or P4 (10 μM) for 8 hours. Intracellular cholesterol was detected by filipin staining (green). Scale bar: 10 μm. (B) PK-15 cells were transfected with NPC1-mCherry plasmid for 24 hours. After pre-treatment of the cells with P4 (10 μM) for 8 hours, the cells were incubated with PRV HN1201 (MOI = 0.1) and P4 (10 μM) at 4°C for 2 hours, followed by a temperature shift to 37°C for 10 minutes to allow entry. After washing with trypsin (1 mg/mL) to remove residual virions on the PM, viral entry was detected by qRT-PCR analysis of viral genome copy numbers in the cells. (C) PK-15 cells were pre-treated with P4 (10 μM) and cholesterol (0, 0.003, 0.01, and 0.03 μg/mL) as indicated for 8 hours. Cells were then incubated with PRV HN1201 (MOI = 0.1), P4 (10 μM), and cholesterol (as indicated) at 4°C for 2 hours, and subsequently shifted to 37°C for 10 minutes to allow entry. Following trypsin wash (1mg/mL) to remove residual virions on the PM, viral entry was detected by qRT-PCR analysis of viral genome copy numbers in the cells. (D) PK-15 cells were treated with vehicle or P4 (10 μM) for 8 hours. The intracellular $Ca^{2+}$ level was detected using Fluo-3 staining (5 μM, green). LAMP1 (red), labeled by immunofluorescence, indicated lysosomes. Scale bar: 10 μm. (E) PK-15 cells were pre-treated with vehicle or P4 (10 μM) for 8 hours and then treated with GPN (200 μM) to release lysosomal $Ca^{2+}$. Intracellular $Ca^{2+}$ levels were detected using Fluo-3 staining (5 μM). (F) PK-15 cells were transfected with an NPC1-mCherry plasmid (red) for 24 hours, followed by treatment with vehicle or P4 (10 μM) for 8 hours. Intracellular $Ca^{2+}$ was detected by Fluo-3 staining (5 μM, green). Scale bar: 10 μm. (G) PK-15 cells were mock-infected, infected with PRV HN1201 (MOI = 0.1), or infected with PRV HN1201 (MOI = 0.1) combined with P4 (10 μM) for 8 hours. Intracellular $Ca^{2+}$ was detected by Fluo-3 staining (5 μM, green). DiI (20 μM, red) indicated the PM. Scale bar: 10 μm. (H) Quantification of the ratio of PM to total Fluo-3 intensity from (G) (n = 10). (I) PK-15 cells were treated with vehicle or P4 (10 μM) for 8 hours. Cells were then mock-infected or infected with PRV HN1201 (MOI = 0.1) as indicated. Intracellular $Ca^{2+}$ was immediately detected by Fluo-3 staining (5 μM). (J) PK-15 cells were treated with vehicle, GPN (200 μM), Baf A1 (1 μM), CPA (20 μM), and CCCP (10 μM) to release organelle-specific $Ca^{2+}$. Subsequently, the cells were infected with PRV (MOI = 0.1). Intracellular $Ca^{2+}$ was immediately detected by Fluo-3 staining (5 μM). (K) PK-15 cells were treated with vehicle, P4 (10 μM), or BAPTA (5 μM) for 8 hours. The cells were incubated with PRV HN1201 (MOI = 0.1), P4 (10 μM), and BAPTA (5 μM) as indicated at 4°C for 2 hours, and then temperature-shifted to 37°C for 10 minutes to allow entry. After washing with trypsin (1 mg/mL) to remove residual virions on the PM, viral entry was detected by qRT-PCR analysis of viral genome copy numbers in

the cells. (L) PK-15 cells were treated with vehicle or P4 (10 μM) for 8 hours. The cells were incubated with PRV HN1201 (MOI = 0.1) and P4 (10 μM) as indicated at 4˚C for 2 hours, then shifted to 37˚C for 0–20 minutes to allow entry. p-DNM1 and DNM1 were analyzed by immunoblotting. (M) PK-15 cells were treated with P4 (10 μM), with or without cholesterol (0.03 μg/mL), $CaCl_2$ (10 mM), or a combination of cholesterol (0.03 μg/mL) and $CaCl_2$ (10 mM) for 8 hours. The cells were incubated with PRV HN1201 (MOI = 0.1) and the indicated compounds at 4˚C for 2 hours, then shifted to 37˚C for 10 minutes to allow entry. After trypsinization (1 mg/mL) to remove residual virions on the PM, viral entry was detected by qRT-PCR analysis of viral genome copy numbers in the cells. (N) PK-15 cells were infected with PRV (MOI = 0.1) and treated with P4 (10 μM), either with or without cholesterol (0.03 μg/mL), $CaCl_2$ (10 mM), or a combination of cholesterol (0.03 μg/mL) and $CaCl_2$ (10 mM) for 24 hours. The viral titer was determined by $TCID_{50}$ assay. Data are expressed as the mean ± SD of 3 independent experiments. *P*-values were determined by Student's *t*-test. $P < 0.05$ was considered statistically significant.

$Ca^{2+}$ efflux channels and thereby stimulate $Ca^{2+}$ release. We observed that the mRNA and protein levels of TPCN1 and TPCN2 remained unchanged in response to PRV infection (S5A–S5D Fig). However, MCOLN1, which encodes the TRPML1 protein, was transcriptionally activated by PRV infection (Fig 5A and 5B). TRPML1 protein levels also increased in PRV-infected cells, as well as in murine uterus (Fig 5C and 5D). Perinuclear lysosome redistribution was seen in cells overexpressing TRPML1-L106P (a mutant associated with mucolipidosis type IV, an autosomal recessive lysosomal storage disorder) [29], TPCN1, and TPCN2, but not with wild-type TRPML1, when treated with P4 (S5E and S5F Fig). TRPML1-Flag expression restored viral replication in P4-treated cells to levels comparable to those in vehicle-treated cells (Fig 5E). During P4 treatment, PRV infection induced $Ca^{2+}$ transport to the PM in cells expressing TRPML1, but not in cells expressing the TRPML1-L106P mutant (Fig 5F). TRPML1-L106P, TPCN1, and TPCN2 were unable to release accumulated $Ca^{2+}$ and cholesterol from lysosomes, while wild-type TRPML1 could (S5G Fig). These results suggest that PRV infection upregulates TRPML1, promoting lysosomal $Ca^{2+}$ and cholesterol release.

To further confirm the role of TRPML1 in PRV infection, TRPML1 expression was interfered using short hairpin RNA (shRNA)-mediated RNA interference (RNAi) (S5H Fig). TRPML1 knockdown inhibited viral entry and replication in both vehicle- and P4-treated cells, as indicated by qRT-PCR analysis of the viral genome and viral titer assays (Fig 5G and 5H). The P4-induced accumulation of $Ca^{2+}$ and cholesterol was reversed by the TRPML1 activator ML-SA1, while the TRPML1 inhibitor ML-SI1 was ineffective (S5I Fig). ML-SI1 inhibited, and ML-SA1 enhanced, viral entry in control cells (S5J Fig). Activation of TRPML1 by ML-SA1 rescued viral entry in P4-treated cells (S5J Fig). DNM1 phosphorylation was increased in response to ML-SI1 and reduced following ML-SA1 treatment (S5K Fig). TRPML1-WT and TRPML1-L106P were expressed in $Mcoln1^{+/+}$ and $Mcoln1^{-/-}$ mouse embryonic fibroblast (MEF) cells (S5L Fig), and viral entry and replication were examined. Comparable levels of viral entry and replication were detected in TRPML1-WT-expressing $Mcoln1^{+/+}$ and $Mcoln1^{-/-}$ MEF cells treated with either vehicle or P4 (Fig 5I and 5J). Viral entry and replication were inhibited in $Mcoln1^{+/+}$ and $Mcoln1^{-/-}$ MEF cells transfected with TRPML1-L106P (Fig 5I and 5J). All these results suggest that viral infection upregulates TRPML1 expression to facilitate viral entry.

## PRV infection promotes lysosomal phosphatidylinositol 3,5-bisphosphate (PI(3,5)P₂) synthesis

Although it has been observed that viral infection enhances TRPML1 expression, TRPML1 activation requires its endogenous activator, $PI(3,5)P_2$ [30]. Immunofluorescence analysis indicated that $PI(3,5)P_2$ levels were increased in PRV-infected cells (Fig 6A and 6B). PIKfyve catalyzes the conversion of PI(3)P to $PI(3,5)P_2$, which is then further converted to PI(5)P by MTM1. $PI(3,5)P_2$ can also be catalyzed back to PI(3)P by Fig 4 (S6A Fig). Immunoblotting and qRT-PCR analysis indicated that PRV infection upregulated PIKfyve and downregulated MTM1 and Fig 4, suggesting that it promoted $PI(3,5)P_2$ synthesis (Figs 6C and S6B–S6D). To

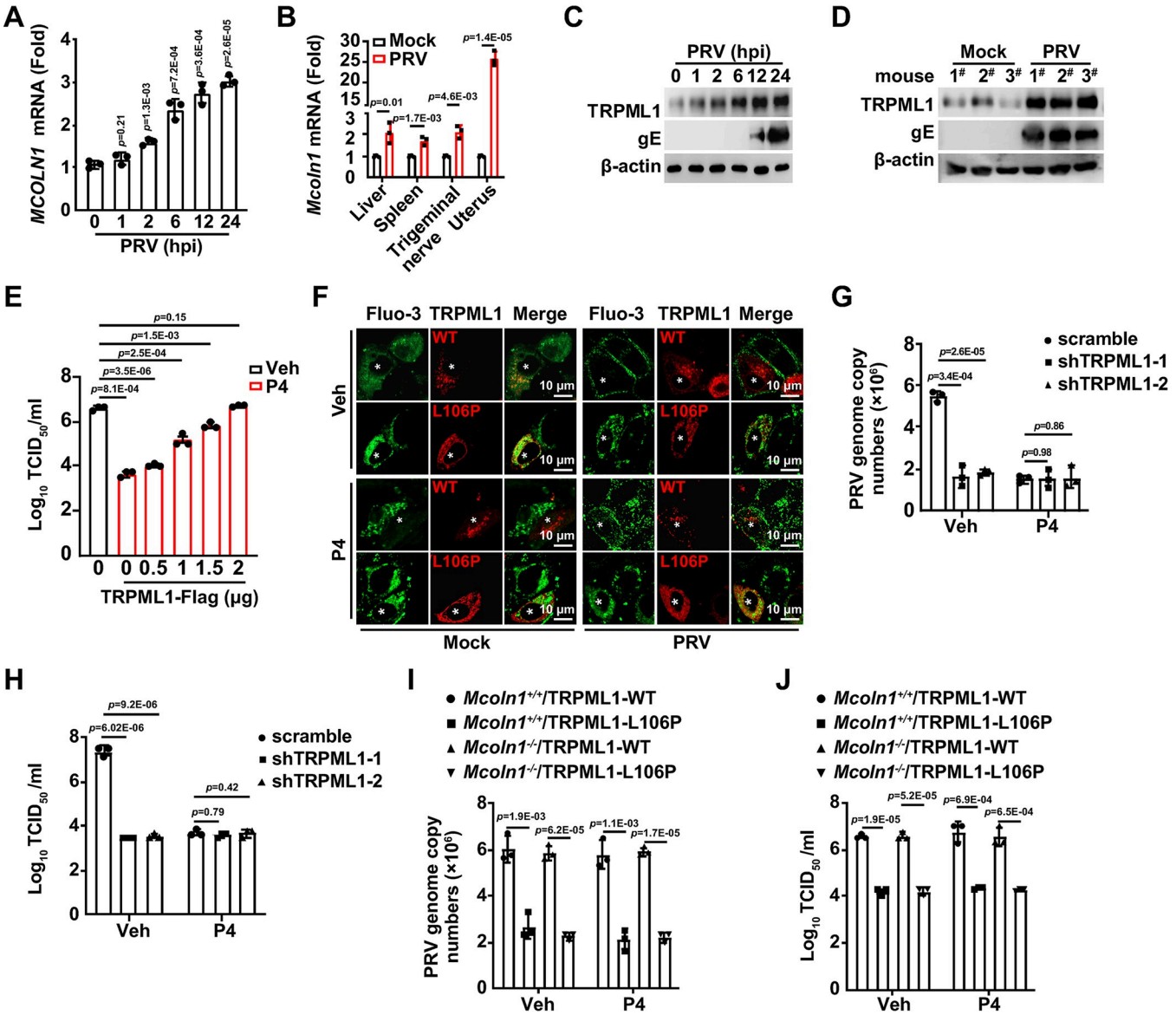

**Fig 5. PRV infection upregulates TRPML1 expression.** (A) PK-15 cells were infected with PRV HN1201 (MOI = 0.1) for 0–24 hours. *MCOLN1* mRNA levels were detected using qRT-PCR analysis. (B) Mice were mock-infected or intranasally infected with PRV HN1201 ($2 \times 10^3$ TCID$_{50}$ per mouse) for two days. *Mcoln1* mRNA levels in the liver, spleen, trigeminal nerve, and uterus were detected using qRT-PCR analysis (n = 3). (C) PK-15 cells were infected with PRV HN1201 (MOI = 0.1) for 0–24 hours. TRPML1 and PRV gE were detected by immunoblotting. (D) Mice were infected with PRV HN1201 as indicated in (B). TRPML1 and PRV gE in the uterus were detected by immunoblotting. (E) PK-15 cells were transfected with a TRPML1-Flag plasmid (0–2 μg) for 24 hours. The cells were then infected with PRV HN1201 (MOI = 0.1) and treated with vehicle or P4 (10 μM) for another 24 hours. The viral titer was determined by the TCID$_{50}$ assay. (F) PK-15 cells were transfected with TRPML1-mCherry variants (WT and L106P, red) for 24 hours. The cells were then either mock-infected or infected with PRV HN1201 (MOI = 0.1) and treated with vehicle or P4 (10 μM) for 8 hours. Intracellular Ca$^{2+}$ was detected by Fluo-3 AM staining (5 μM, green). Scale bar: 10 μm. (G) Scramble and shTRPML1 PK-15 cells were treated with vehicle or P4 (10 μM) for 8 hours. The cells were incubated with PRV HN1201 (MOI = 0.1), vehicle, or P4 (10 μM) as indicated, at 4°C for 2 hours and then temperature-shifted to 37°C for 10 minutes to allow entry. After washing with trypsin (1 mg/mL) to remove residual virions on the PM, viral entry was detected by qRT-PCR analysis of viral genome copy numbers in the cells. (H) Scramble and shTRPML1 PK-15 cells were infected with PRV HN1201 (MOI = 0.1) and treated with vehicle or P4 (10 μM) for 24 hours. The viral titer was determined by the TCID$_{50}$ assay. (I) *Mcoln1*$^{+/+}$ and *Mcoln1*$^{-/-}$ MEF cells were transfected with TRPML1-WT or TRPML1-L106P for 24 hours. The cells were incubated with PRV HN1201 (MOI = 0.1), vehicle, or P4 (10 μM) as indicated, at 4°C for 2 hours, and then temperature-shifted to 37°C for 10 minutes to allow entry. After washing with trypsin (1 mg/mL) to remove residual virions on the PM, viral entry was detected by qRT-PCR analysis of viral genome copy numbers in the cells. (J) *Mcoln1*$^{+/+}$ and *Mcoln1*$^{-/-}$ MEF cells were transfected with TRPML1-WT or TRPML1-L106P for 24 hours. The cells were then infected with PRV HN1201 (MOI = 0.1) and treated with vehicle or P4 (10 μM) for another 24 hours. The viral titer was determined by the TCID$_{50}$ assay. Data are expressed as the mean ± SD of 3 independent experiments. *P*-values were determined by Student's *t*-test. *P* < 0.05 was considered statistically significant.

determine the lysosomal PI(3,5)P2 content, lysosomes from both mock- and PRV-infected cells were purified (Figs 6C and S6E). Lipid dot blot analysis showed that the lysosomal PI(3,5)$P_2$ level was increased due to PRV infection (Fig 6D). The PIKfyve inhibitor YM-201636 prevented the PRV-induced upregulation of lysosomal PI(3,5)$P_2$ (Fig 6D). Co-localization of PI(3,5)$P_2$ with LAMP1 was enhanced by PRV infection, as indicated by the results of immunofluorescence analysis (Fig 6E and 6F). These results suggest that viral infection promotes lysosomal PI(3,5)$P_2$ synthesis.

Lysosomal $Ca^{2+}$ release was examined next. No increased $Ca^{2+}$ signal was detected in cells treated with the PIKfyve inhibitor YM-201636, the ML-SI1, or the PI(3,5)$P_2$ chelator poly-lysine [31] upon PRV infection, suggesting that PI(3,5)$P_2$ is essential for TRPML1 activation (Fig 6G and 6H). Chelating PI(3,5)$P_2$ with poly-lysine significantly affected viral entry (Fig 6I). To further confirm that PI(3,5)$P_2$ is responsible for the PRV-induced activation of TRPML1, an inducible FKBP-FRB heterodimerization system was used to deplete PI(3,5)$P_2$ on the lysosomes (Fig 6J). FKBP was targeted to lysosomes by fusion with Rab7, and MTM1 was retained in the cytoplasm, fused with mCherry-FRB. The application of the chemical inducer rapamycin led to the recruitment of mCherry-FRB-MTM1 to lysosome membranes by binding BFP-FKBP-Rab7 [32], which rapidly and irreversibly converted PI(3,5)$P_2$ to PI/PI(5)P (Fig 6J and 6K). Rapamycin treatment caused $Ca^{2+}$ accumulation in the lysosomes (Fig 6L). Rapamycin abrogated the PRV-induced $Ca^{2+}$ release in cells expressing both BFP-FKBP-Rab7 and mCherry-FRB-MTM1 (Fig 6M). It has been demonstrated that the TRPML1-7Q mutant, with seven substitutions (R42Q/R43Q/R44Q/K55Q/R57Q/R61Q/K62Q), is unable to bind to PI(3,5)P2 [30]. PRV stimulated $Ca^{2+}$ release in *Mcoln1*-null MEF cells expressing TRPML1-WT (Fig 6N). Expression of TRPML1-7Q in either control or *Mcoln1*$^{-/-}$ MEF cells prevented PRV-induced $Ca^{2+}$ release (Fig 6N). Collectively, these results suggest that viral infection promotes lysosomal PI(3,5)$P_2$ synthesis to activate TRPML1.

## Murine double minute 2 (MDM2) is the E3 ubiquitin ligase of TRPML1

To determine whether P4 influenced TRPML1 expression, we evaluated TRPML1 expression during P4 treatment using both qRT-PCR and immunoblotting analysis. TRPML1 mRNA levels were unaltered by P4 treatment (S7A Fig). Notably, TRPML1 protein levels decreased in cells treated with P4 and in the murine uterus following P4 injection, suggesting that P4 affects TRPML1 expression at the post-translational level (S7B–S7D Fig). Additionally, P4 abolished PRV-enhanced TRPML1 expression (Fig 7A). These results suggest that viral infection may decrease P4 levels to prevent TRPML1 degradation.

The ubiquitin-proteasome system and autophagy are two major intracellular protein quality control pathways [33, 34]. The proteasome was inhibited by MG-132, while autophagy was inhibited by chloroquine, to determine the pathway responsible for P4-induced TRPML1 degradation. Immunoblotting analysis indicated that MG-132, but not chloroquine, blocked TRPML1 degradation upon P4 treatment (S7E Fig). This finding was further verified by the cycloheximide (CHX) chase assay (Fig 7B). TRPML1 ubiquitination was also examined. Compared to mock-infected cells, PRV-infected cells showed decreased TRPML1 ubiquitination (Fig 7C). P4 treatment enhanced TRPML1 ubiquitination in challenged cells (Figs 7C and S7F). To determine the ubiquitin linkage type on TRPML1, we used ubiquitin mutants: WT, K48-only (with all other lysines mutated to arginines), or K63-only (with all other lysines mutated to arginines). TRPML1 was found to have both K48- and K63-linked polyubiquitin chains, suggesting that K48/K63-linked polyubiquitination occurred (Fig 7D). Given that polyubiquitin chains are covalently attached to the lysine residues on the substrate [35], we aimed to identify which lysine residues in TRPML1 were modified by polyubiquitin chains. We

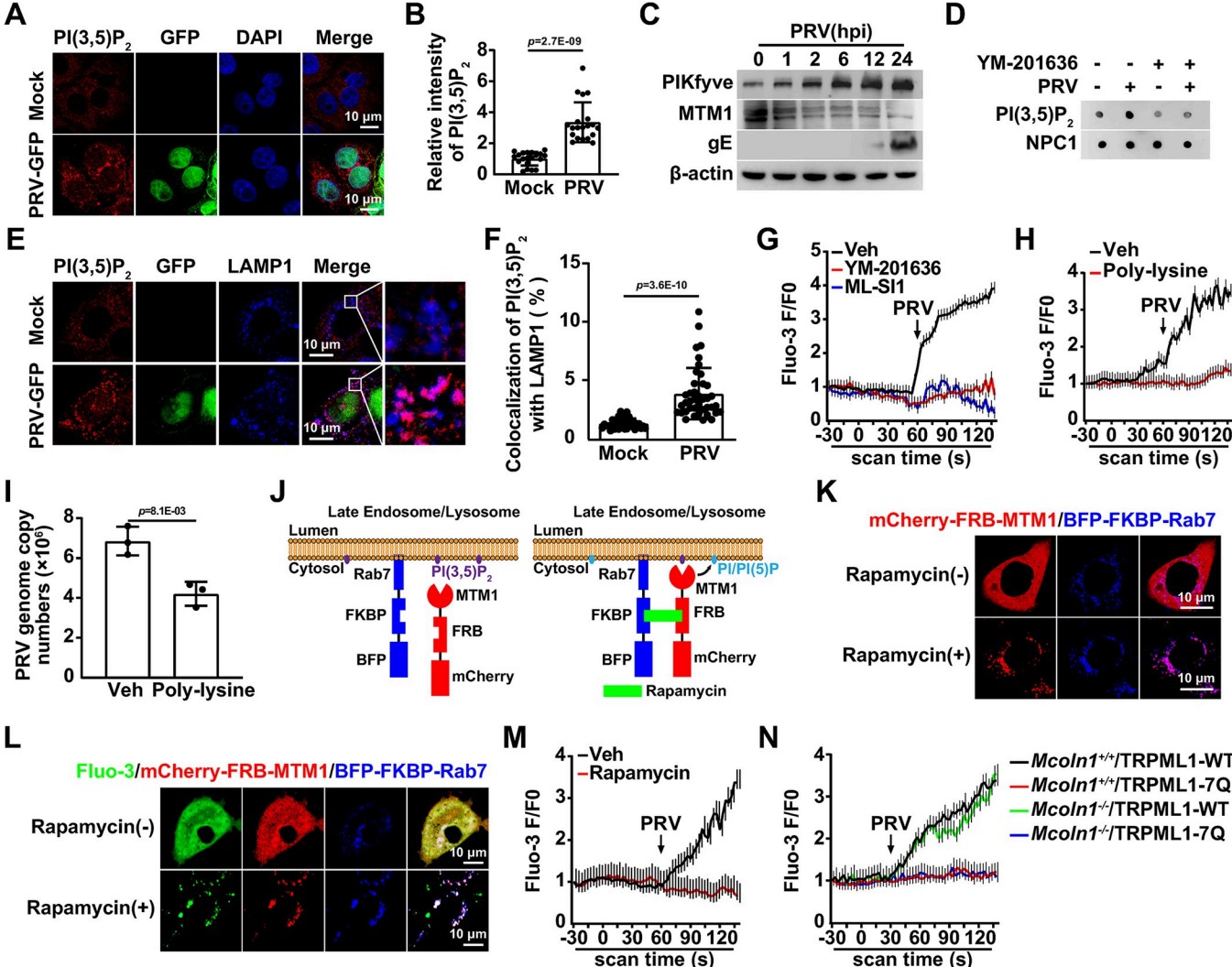

**Fig 6. PRV infection enhances lysosomal PI(3,5)P$_2$ synthesis.** (A) PK-15 cells were mock-infected or infected with PRV-GFP (MOI = 0.1) for 24 hours. PI(3,5)P$_2$ (red) was detected by immunofluorescence analysis. Scale bar: 10 μm. (B) Quantification of the relative intensity of PI(3,5)P$_2$ from (A) (n = 20). (C) PK-15 cells were infected with PRV HN1201 (MOI = 0.1) for 0–24 hours. PIKfyve, MTM1, and PRV gE were detected by immunoblotting. (D) PK-15 cells were mock-infected or infected with PRV HN1201 (MOI = 0.1) and treated with YM-201636 (10 μM) for 24 hours. PI(3,5)P$_2$ in the lysosomal fraction was detected by lipid dot-blot analysis. The fraction was analyzed by immunoblotting with an antibody against NPC1 (lysosome marker). (E) PK-15 cells were mock-infected or infected with PRV-GFP (MOI = 0.1) for 24 hours. The colocalization of PI(3,5)P$_2$ (red) with LAMP1 (lysosomal marker, blue) was detected by immunofluorescence analysis. Scale bar: 10 μm. (F) Quantification of colocalization of PI(3,5)P$_2$ with LAMP1 from (E) (n = 40). (G) PK-15 cells were treated with vehicle, YM-201636 (10 μM), or ML-SI1 (10 μM) for 8 hours. Cells were then infected with PRV HN1201 (MOI = 0.1). Intracellular Ca$^{2+}$ levels were immediately detected using Fluo-3 staining (5 μM). (H) PK-15 cells were treated with vehicle or poly-lysine (50 μg/mL) for 8 hours. Cells were then infected with PRV HN1201 (MOI = 0.1). Intracellular Ca$^{2+}$ levels were immediately detected using Fluo-3 staining (5 μM). (I) PK-15 cells were treated with vehicle or poly-lysine (50 μg/mL) for 8 hours. Cells were incubated with PRV HN1201 (MOI = 0.1), vehicle, or poly-lysine (50 μg/mL) as indicated at 4°C for 2 hours, and then temperature-shifted to 37°C for 10 minutes to allow entry. After washing with trypsin (1 mg/mL) to remove residual virions on the PM, viral entry was detected by qRT-PCR analysis of viral genome copy numbers in the cells. (J) Schematic representation of the rapamycin-inducible heterodimerization system used to recruit MTM1 to the lysosomal membrane. Rab7 is a lysosome-specific Rab protein. MTM1 is a PI 3 phosphatase that can convert PI(3,5)P$_2$ and PI(3)P into PI(5)P and PI, respectively. (K) HeLa cells were cotransfected with mCherry-FRB-MTM1 (red) and BFP-FKBP-Rab7 (blue) for 24 hours. Cells were then treated with rapamycin (500 nM) for 20 minutes. The subcellular localization of mCherry-FRB-MTM1 and BFP-FKBP-Rab7 was observed by fluorescence microscopy. Scale bar: 10 μm. (L) HeLa cells were transfected with mCherry-FRB-MTM1 (red) and BFP-FKBP-Rab7 (blue) for 24 hours. Cells were then treated with rapamycin (500 nM) for 20 minutes. Intracellular Ca$^{2+}$ levels were detected using Fluo-3 staining (5 μM). Scale bar: 10 μm. (M) HeLa cells were transfected with mCherry-FRB-MTM1 (red) and BFP-FKBP-Rab7 (blue) for 24 hours. Cells were then treated with rapamycin (500 nM) for 20 minutes and infected with PRV HN1201 (MOI = 0.1). Intracellular Ca$^{2+}$ levels were immediately detected using Fluo-3 staining (5 μM). (N) *Mcoln1*$^{+/+}$ and *Mcoln1*$^{-/-}$ MEF cells were transfected with TRPML1-WT or TRPML1-7Q for 24 hours. Cells were then infected with PRV HN1201 (MOI = 0.1). Intracellular Ca$^{2+}$ levels were immediately detected using Fluo-3 staining (5 μM). Data are expressed as the mean ± SD of 3 independent experiments. *P*-values were determined by Student's *t*-test. *P* < 0.05 was considered statistically significant.

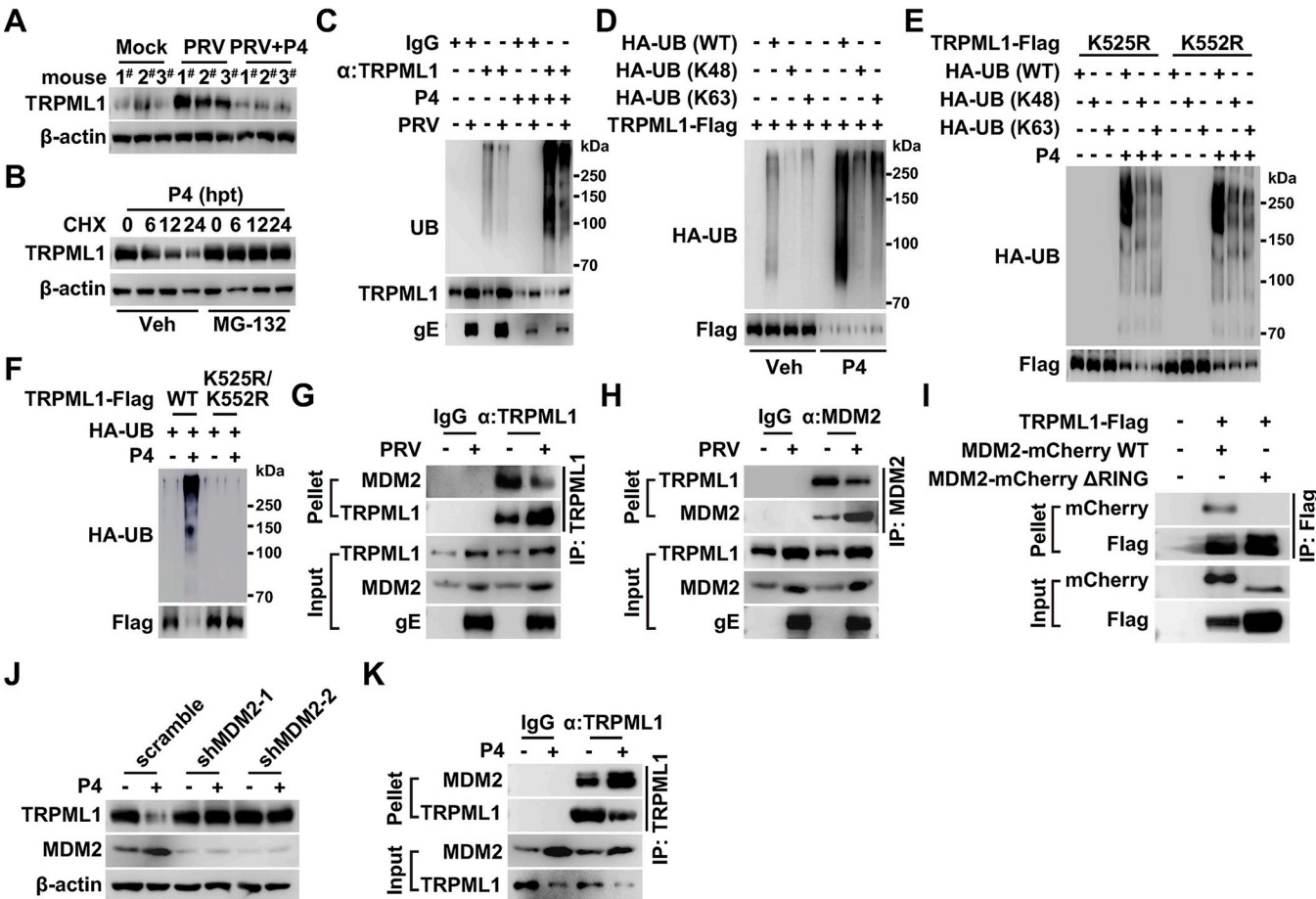

**Fig 7. MDM2 is the E3 ubiquitin ligase of TRPML1.** (A) Mice were either mock-infected or intranasally infected with PRV HN1201 ($2 \times 10^3$ TCID$_{50}$ per mouse) and were daily injected with vehicle or P4 (10 mg/kg per mouse) for two days. TRPML1 in the uterus was detected by immunoblotting (n = 3). (B) PK-15 cells were treated with P4 (10 μM) and MG-132 (10 μM) as indicated for 0–24 hours. TRPML1 degradation was analyzed by CHX assay for the same duration. (C) PK-15 cells were infected with PRV HN1201 (MOI = 0.1) and treated with P4 (10 μM) for 14 hours as indicated. TRPML1 ubiquitination was analyzed by ubiquitination assay. (D) HEK293T cells were transfected with TRPML1-Flag and HA-UB variants (WT, K48, and K63) as indicated for 24 hours. Cells were then treated with vehicle or P4 (10 μM) for 8 hours. TRPML1-Flag ubiquitination was analyzed by ubiquitination assay. (E) HEK293T cells were transfected with TRPML1-Flag mutants (K525R and K552R) and HA-UB variants (WT, K48, and K63) as indicated for 24 hours. Cells were then treated with vehicle or P4 (10 μM) for 8 hours. Ubiquitination of TRPML1-Flag (K525R and K552R) was analyzed by ubiquitination assay. (F) HEK293T cells were transfected with TRPML1-Flag variants (WT and RR) and HA-UB as indicated for 24 hours. Cells were then treated with vehicle or P4 (10 μM) for 8 hours. TRPML1-Flag (WT and RR) ubiquitination was analyzed by ubiquitination assay. (G) PK-15 cells were either mock-infected or infected with PRV HN1201 (MOI = 0.1) for 24 hours. The interaction between TRPML1 and MDM2 was analyzed by CoIP analysis. (H) PK-15 cells were infected with PRV HN1201 as indicated in (G). The interaction between MDM2 and TRPML1 was analyzed by CoIP analysis. (I) HEK293T cells were transfected with TRPML1-Flag and MDM2-mCherry variants (WT and ΔRING) for 24 hours. The interaction of TRPML1-Flag with MDM2-mCherry variants was analyzed by CoIP analysis. (J) Scramble and shMDM2 PK-15 cells were treated with vehicle or P4 (10 μM) for 24 hours. TRPML1 and MDM2 levels were analyzed by immunoblotting. (K) PK-15 cells were treated with vehicle or P4 (10 μM) as indicated for 24 hours. The interaction between TRPML1 and MDM2 was analyzed by CoIP analysis.

generated and transfected ten TRPML1-Flag mutant plasmids into cells: TRPML1-Flag K46R, K55R, K59R, K62R, K65R, K72R, K337R, K410R, K525R, and K552R. Among these, the TRPML1-Flag K525R and K552R mutants exhibited decreased ubiquitination following P4 treatment, whereas ubiquitination of the other mutants was comparable to the wild-type control level (Figs 7E and S7G). Subsequently, the K525 and K552 lysine residues were simultaneously mutated to arginine, creating the TRPML1-Flag K525R/K552R double mutant. This double mutant showed no detectable ubiquitination after P4 treatment (Fig 7F). These results suggest that P4 induces K48/K63-linked polyubiquitination of TRPML1 at K525 and K552.

E3 ubiquitin ligases catalyze the conjugation of polyubiquitin chains onto substrates [36]. To identify the E3 ubiquitin ligase responsible for TRPML1 ubiquitination, co-immunoprecipitation (CoIP) coupled with tandem mass spectrometry was performed. Three potential E3 ubiquitin ligases, including MDM2, tripartite motif containing 21 (TRIM21), and ring finger and CCCH-type domains 2 (RC3H2), were found to associate with TRPML1 (S7H Fig). Their interaction with TRPML1 was further verified. CoIP analysis revealed that only MDM2 interacted with TRPML1, while TRIM21 and RC3H2 did not (S7I Fig). Additional results showed that TRPML1 and MDM2 interacted within the cells (Fig 7G and 7H). Notably, PRV infection attenuated the interaction between TRPML1 and MDM2, suggesting that the viral infection impeded their interactions to prevent TRPML1 ubiquitination (Fig 7G and 7H). MDM2 is a RING finger-dependent E3 ubiquitin ligase [37]. Deletion of the RING domain in MDM2 (MDM2-mCherry ΔRING) abolished the interaction between TRPML1 and MDM2 (Fig 7I).

MDM2 was then knocked down, which allowed us to examine TRPML1 degradation. Immunoblot analysis indicated that P4-induced TRPML1 degradation was completely blocked in cells with MDM2 knockdown (Fig 7J), further confirming that MDM2 acts as the E3 ubiquitin ligase for TRPML1. To determine the role of P4 in the interaction between TRPML1 and MDM2, MDM2 expression in response to P4 treatment was investigated. Intriguingly, P4 significantly upregulated MDM2 expression both in the cell culture model and in murine uterus (S7J and S7K Fig). CoIP analysis demonstrated that P4 enhanced the interaction between TRPML1 and MDM2 (Fig 7K). Taken together, these results indicated that viral infection inhibited the interaction between TRPML1 and MDM2, leading to the stabilization of TRPML1, an effect that was counteracted by P4.

## Discussion

Miscarriage may follow infection with herpesvirus, such as equid herpesvirus-1 [38], bovine herpesvirus-1 [39], caprine herpesvirus-1 [40], and canine herpesvirus [41], as well as PRV [42] in domestic animals [43]. The present study demonstrated that intranasal infection of mice with PRV affected HPOA-regulated P4 levels and subsequent pregnancy rates. We further discovered that P4 inhibited lysosome-dependent viral entry through stimulation of proteasomal TRPML degradation (Fig 8). This is a novel mechanism of how herpesvirus causes infertility and miscarriage, shedding light on the prevention of herpesviral abortion in domestic animals and humans.

It has been reported that P4 ameliorates the severity of SARS-CoV-2-caused pneumonia in the Syrian hamster model [44]. Stimulation of the PGR by P4 activates the tyrosine kinase SRC, which in turn phosphorylates the transcriptional factor interferon regulatory factor 3 at Y107, leading to its activation and the induction of antiviral genes [45]. These findings suggest that P4 exhibits antiviral activity. SARS-CoV-2-infected patients have been shown to exhibit increased P4 levels, which correlate with decreased COVID-19 severity [45]. Although P4 also inhibited PRV infection, it was observed that PRV infection decreased P4 levels, which was beneficial for viral replication. Moreover, P4 prevented immunostimulant-activated innate immunity, suggesting that P4 exhibits an immunosuppressive effect [23]. We demonstrated that the PGR is not required for the P4-mediated inhibition of PRV infection. Collectively, these data indicate that the role of P4 in antiviral activity varies across different viral infections, and that modulating P4 levels could be a potent therapeutic strategy against viral infections.

Lysosomal ion channels are involved in viral infections [46]. Our data indicate that PRV infection transcriptionally upregulates TRPML1 mRNA expression both in vitro and in vivo. Palmieri et al. used chromatin immunoprecipitation sequencing to identify direct transcription factor EB (TFEB) targets and found that TRPML1 expression is regulated by TFEB [47].

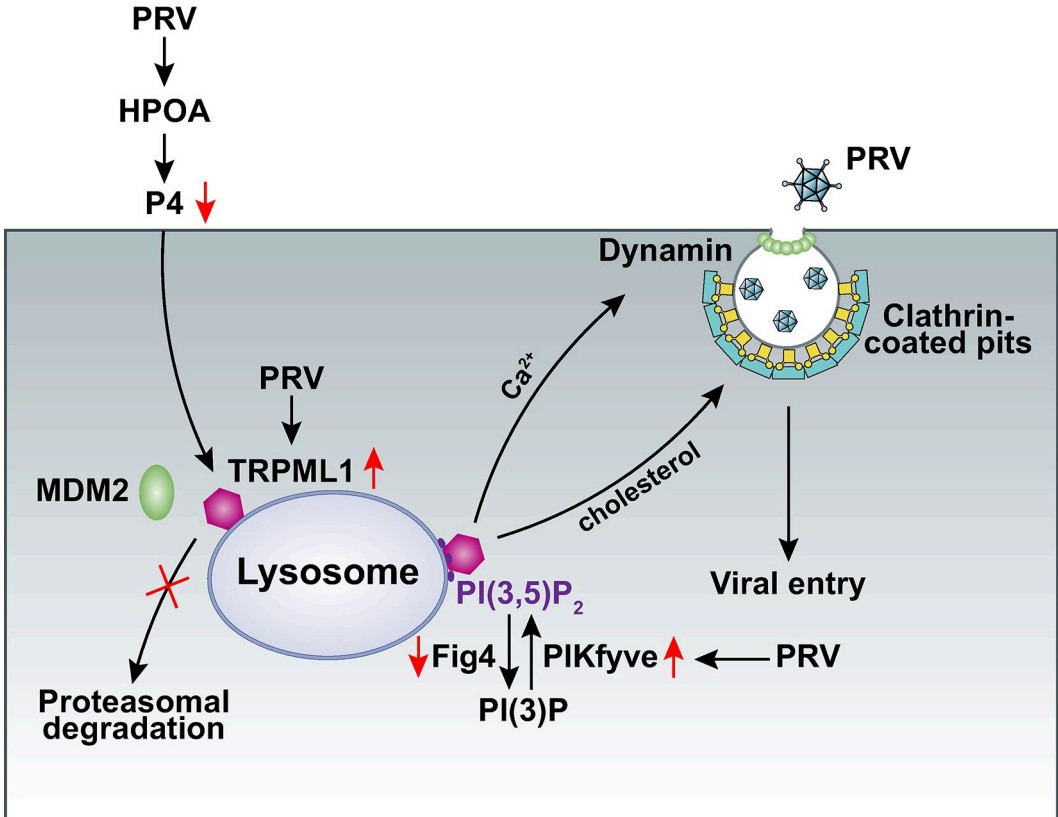

**Fig 8. A schematic model showing PRV evades P4-mediated antiviral defense and impairs female fertility.**

Given that PRV infection inhibits autophagy in permissive cells *in vitro* [48], and considering that TFEB plays a pivotal role in autophagy regulation [49], we speculate that PRV infection might not activate *MCOLN1* transcription through TFEB. Additionally, we demonstrated that TRPML1 can be post-translationally regulated by the E3 ubiquitin ligase MDM2. Due to the lack of enough studies on the regulation of TRPML1 expression, the function of TRPML in lysosomes needs to be explored further.

Viruses manipulate cellular lipids and membranes at each stage of their life cycle [50]. PI$(3,5)P_2$ is required for the entry of filoviruses, including Ebola virus [51, 52]. Conversely, PRV infection enhances lysosomal PI$(3,5)P_2$ synthesis to activate TRPML1. Untargeted liquid chromatography-mass spectrometry-based metabolomics revealed that lipids accounted for over 50% of the altered metabolites, including glycerophospholipids, sphingolipids, glycerolipids, and fatty acyls [53]. It has also been shown that PRV infection promotes lipid synthesis [54], suggesting that the virus reprograms lipid metabolism to activate TRPML1 for viral entry. Furthermore, PRV infection specifically results in the translocation of lysosomal cholesterol and $Ca^{2+}$ to the PM for clathrin-mediated endocytosis, which is pivotal for viral entry. Whether other viruses, such as filoviruses and Ebola virus, use the same mechanism for entry requires further investigation.

Lysosomes serve as a metabolic signaling hub in cells, and perturbed lysosomal function results in LSDs [55]. Mucolipidosis type IV is an inherited disorder characterized by delayed development and vision impairment, caused by mutations in *MCOLN1*. Our data indicate that P4 induces LSDs via MDM2-mediated proteasomal degradation of TRPML1. Although it has been suggested that P4 prevents neurodegeneration [56], the potential for P4-induced LSDs

should be considered in clinical trials. Niemann-Pick disease is another neurodegenerative LSD, caused by mutations in NPC1 [57]. NPC1 mutant cells have a considerably reduced acidic compartment $Ca^{2+}$ store compared to wild-type cells. Moreover, chelating luminal endocytic $Ca^{2+}$ in normal cells with high-affinity Rhod-dextran induces a cellular phenotype reminiscent of NPC disease [58]. Prevention of lysosomal $Ca^{2+}$ efflux through TRPML1 results in cholesterol accumulation in the lysosomes, whereas NPC1 overexpression rescues this effect. These data suggest that disturbances in TRPML1 affect NPC1 function. Nevertheless, $Ca^{2+}$ accumulation and lysosomal function cannot be restored by NPC1 overexpression in P4-treated cells, suggesting that the role of $Ca^{2+}$ in lysosomal function regulation is more crucial than that of cholesterol. How lysosomes are regulated by $Ca^{2+}$ and cholesterol remains to be elucidated.

## Materials and methods

### Ethics statement

Experiments involving animals were approved by the Committee on the Ethics of Animal Care and Use of Henan Agricultural University (HNND2019031004). The study was conducted in accordance with the Guide for the Care and Use of Animals in Research of the People's Republic of China.

### Mice

The present study utilized 6–8-week-old C57BL/6J mice. *Mcoln1*[+/+] and *Mcoln1*[-/-] (C57BL/6JSmoc-*Mcoln1*[em1Smoc]) mice on a C57BL/6J background were purchased from SHANGHAI MODEL ORGANISMS. Genotyping was performed using PCR according to the manufacturer protocol. Mice were housed in specific pathogen-free facilities with a 12-h light-dark cycle and a controlled temperature of 22˚C. Animal protocols were in accordance with the guide for the care and use of laboratory animals and related ethical regulations at Henan Agricultural University. Mice were injected with leuprolide (1.5 mg/kg) once a week for two weeks or with P4 (10 mg/kg per mouse) every day for 10 days.

### Cells

PK-15, HeLa, HEK293T, and MEF cells (derived from 12.5-day murine embryos) were maintained in DMEM supplemented with 10% fetal bovine serum (FBS), 1% penicillin/streptomycin, and 1% glutamine at 37˚C with 5% $CO_2$.

### Viruses

PRV HN1201 and PRV-GFP were used as previously described [12]. Viruses were propagated and tittered using a 50% tissue culture infective dose ($TCID_{50}$) assay in Vero cells. For cell-based assays, cells were infected with PRV (MOI = 0.1, MOI = 10, or MOI = 100, as indicated). EdU-labeled PRV HN1201 was used as previously described [59]. For *in vivo* studies, mice were intranasally infected with PRV HN1201 ($2 \times 10^2$ $TCID_{50}$ per mouse or $2 \times 10^3$ $TCID_{50}$ per mouse as indicated).

### Materials and plasmids

Leuprolide acetate was purchased from Selleck. P4, rapamycin, chloroquine, MG-132, YM-201636, CHX, BAPTA, CCCP, ML-SA1, and ML-SI1 were purchased from MedChemExpress. Baf-A1 and CPA were purchased from GLPBIQ. GPN was purchased from Abcam. Chicago

Sky Blue 6B, HT-DNA, and LPS were purchased from Sigma-Aldrich. Poly(I:C) and poly(dA:dT) were purchased from InvivoGen.

Anti-AP2B1, anti-NPC1, anti-DNM1, anti-UB, anti-PIkfyve, anti-MTM1, anti-MDM2, anti-TPCN1, anti-Tom20, anti-lamin B1, anti-LAMP1, anti-mCherry and anti-β-actin antibodies were purchased from Proteintech. Anti-Flag and anti-calnexin antibodies were purchased from Sigma-Aldrich. Anti-TRPML1 antibody was purchased from Thermo. Anti-p-DNM1 antibody was purchased from R&D Systems. Anti-HA antibody was purchased from GenScript. Anti-PI(3,5)P2 antibody was purchased from Echelon. Anti-PR antibody was purchased from Servicebio. Anti-TPCN2 antibody was purchased from Zeye. Antiserum against PRV gE was generated by immunization of mice with purified recombinant gE.

The coding sequences for PR, TRPML1, TPCN1, TPCN2, and MDM2 were amplified using cDNA from PK-15 cells and cloned into p3×Flag-CMV-14 (PR-Flag, TRPML1-Flag, TPCN1-Flag, and TPCN2-Flag) or mCherry-N1 (TRPML1-mCherry, TPCN1-mCherry, TPCN2-mCherry, and MDM2-mCherry). Primers used for gene cloning are listed in **S1 Table**. All the mutant constructs (TRPML1-Flag L106P, TRPML1-Flag 7Q [R42Q/R43Q/R44Q/K55Q/R57Q/R61Q/K62Q], TRPML1-mCherry L106P, and MDM2-mCherry ΔRING [aa.1-435]) were generated using a QuikChange Site-Directed Mutagenesis Kit (Angilent) according to the manufacturer instructions. BFP-FKBP-Rab7 and mCherry-FRB-MTM1 were synthesized by GenScript. AP2B1-mCherry, NPC1-Flag, and NPC1-mCherry were used as previously described [12]. HA-UB (WT), HA-UB (K48), and HA-UB (K63) were gifts from Bo Zhong (College of Life Sciences, Wuhan University, China) [60]. DNA construct transfection was performed using Lipofectamine 3000 (Invitrogen) according to the manufacturer instructions.

## Cell viability assay

Cell viability was evaluated using a cell counting kit-8 (CCK-8) according to the manufacturer instructions (Dingguo). Briefly, cells were seeded into 96-well plates at a concentration of $0.8 \times 10^4$ cells per well. On the next day, the medium was changed to DMEM/10% FBS supplemented with P4 (0–30 μM) and the cells were incubated for 12–72 h. The CCK-8 agent (10 μL) was added to each well at specific time points and the cells were incubated at 37°C for 3 h. The absorbance was detected at 450 nm with a microplate reader (VARIOSKAN FLASH, Thermo).

## Immunoblotting analysis

For immunoblotting, cells were harvested and lysed using a lysis buffer (50 mM Tris–HCl pH 8.0, 150 mM NaCl, 1% Triton X-100, 1% sodium deoxycholate, 0.1% SDS, and 2 mM $MgCl_2$) supplemented with a protease and phosphatase inhibitor cocktail (MedChemExpress). Protein samples were separated using SDS-PAGE and then transferred to a membrane (Millipore), which was incubated in 5% nonfat milk for 1 h at room temperature. The membrane was then incubated with a primary antibody overnight at 4°C, followed by horseradish peroxidase-conjugated secondary antibody for 1 h at room temperature. Immunoblotting results were visualized using Luminata Crescendo Western HRP substrate (Millipore) on a GE AI600 imaging system.

## Lysosome purification by iodixanol density gradient centrifugation

Cell homogenates were centrifuged serially at $1,000 \times g$ for 10 min and $20,000 \times g$ for 20 min. The 20,000-g pellets were collected and placed on 8%, 12%, 16%, 19%, 22.5%, and 27% (v/v) iodixanol gradients and then centrifuged at $150,000 \times g$ for 4 h. Fractions (0.8 mL each) were

collected from the top to the bottom. The fraction with the purest lysosome content was 1. Organelle purity was assessed by assaying equal volumes of all fractions for presence of organelle markers.

### Lipid dot blot analysis

Lipid extraction from the lysosomes fraction was performed as described previously [61]. Briefly, methanol 12.1 N HCl (10:1) was added to fraction 1 at a volumetric ratio of 1:1 and chloroform was added at a volumetric ratio of 2:1 (solvent membrane fraction) to facilitate phase separation. The organic phase was subsequently extracted with the addition of methanol 1 N HCl (1:1) at a volumetric ratio of 1:1. The organic phase was $N_2$ dried and the pellets were used for lipid dot blot analysis. Specifically, lipids were spotted onto a Hybond-C nitrocellulose membrane. The membrane was then blocked in 3% fatty acid-free BSA at room temperature for 1 h and probed with mouse anti-PI(3,5)$P_2$ antibody (1:1,000 dilution; Echelon). Bound antibody was detected by immunoblotting.

### Ubiquitination assay

Cells were harvested and lysed in 1 mL of IP buffer (50 mM Tris–HCl pH 7.4, 150 mM NaCl, 1% NP-40, 1% sodium deoxycholate, 5 mM EDTA, and 5 mM EGTA) and clarified by centrifugation at $16,000 \times g$ for 10 min at 4˚C. Next, 900-µL aliquots were incubated with 40 µL of a 1:1 slurry of sepharose conjugated with either anti-TRPML1 or anti-Flag mouse mAb (Sigma-Aldrich) for 4 h at 4˚C. The beads were then washed four times with IP buffer and eluted with SDS sample buffer by boiling for 10 min before immunoblot analysis.

### CoIP and tandem mass spectrometry

Cells were harvested and lysed in 1 mL of CoIP buffer (50 mM Tris–HCl pH 7.4, 150 mM NaCl, 1% NP-40, 5 mM EDTA, and 5 mM EGTA) supplemented with a protease and phosphatase inhibitor cocktail (MedChemExpress) and clarified by centrifugation at $16,000 \times g$ for 10 min at 4˚C. Next, 900-µL aliquots were incubated with 40 µL of a 1:1 slurry of sepharose conjugated with either IgG (GE Healthcare), anti-TRPML1, anti-MDM2, or anti-Flag mouse monoclonal antibody (Sigma-Aldrich) for 4 h at 4˚C. The beads were then washed three times with CoIP buffer and eluted with SDS sample buffer by boiling for 10 min before immunoblotting.

To identify the E3 ubiquitin ligase that was responsible for the ubiquitination of TRPML1, the eluted proteins associated with Sepharose conjugated with anti-TRPML1 antibody were subjected to SDS-PAGE and stained using Coomassie Blue. The visible protein bands were excised and subjected to trypsin digestion overnight. The resulting tryptic peptides were analyzed using an Easy nLC1200 (Thermo), which was directly interfaced with a Q Exactive mass spectrometer (Thermo). The potential E3 ubiquitin ligases that interacted with TRPML1 are represented in S7H Fig.

### qRT-PCR

Total RNA was isolated using TRIzol reagent (TaKaRa) and subjected to cDNA synthesis using a PrimeScript RT reagent kit (TaKaRa). The qRT-PCR was performed in triplicate using SYBR Premix Ex Taq (TaKaRa) according to the manufacturer instructions. Data were normalized to the level of β-actin expression in each individual sample. Melting curve analysis indicated the formation of a single product in all cases. The $2^{-\Delta\Delta Ct}$ method was used to calculate relative expression changes. Primers used for qRT-PCR are listed in **S1 Table**.

## RNAi

Lentivirus-mediated gene silencing was conducted as previously described [62]. Briefly, shRNAs against *TRPML1* and *MDM2* (**S2 Table**) were synthesized as double-strand oligonucleotides, cloned into the pLKO.1 vector, and co-transfected with packaging plasmids pMD2. G and psPAX into HE293T cells. Lentiviruses were harvested 48 h post transfection and used to infect cells that were then selected with puromycin (4 μg/mL) for seven days. Knockdown efficiency was determined using immunoblotting analysis.

## Generation of gene knockout cell lines using CRISPR/Cas9

Lentivirus-mediated gene silencing was conducted as previously described [63]. Briefly, small guide RNAs (**S2 Table**) against *PGR* were synthesized as double-strand oligonucleotides, cloned into the lentiCRISPR v2 vector, and co-transfected with packaging plasmids pMD2.G and psPAX into HE293T cells. Lentiviruses were harvested 48 h post transfection and used to infect cells that were then selected with puromycin (4 μg/mL) for seven days. Single-clonal knockout cells were obtained by serial dilution and verified using Sanger sequencing and immunoblotting analysis.

## Viral attachment assay

Cells were incubated with PRV HN1201 or EdU-labeled PRV HN1201 at 4˚C for 2 h. After three extensive washes with ice-cold phosphate-buffered saline (PBS), viral attachment assays were performed using qRT-PCR analysis of PRV genome copy numbers or by fluorescence analysis of EdU-labeled PRV on the PM.

## Viral entry assay

Cells were incubated with PRV HN1201 or EdU-labeled PRV HN1201 at 4˚C for 2 h. Then, the cells were extensively washed with ice-cold PBS three times and incubated at 37˚C for 10 min to allow entry. After washing with trypsin (1 mg/mL) to remove the residual virions on the PM, viral entry was detected via qRT-PCR analysis of viral genome copy numbers or by fluorescence analysis of EdU-labeled PRV in the cells [12].

## Immunofluorescence and filipin staining

Cells cultured on coverslips in 12-well plates were fixed with 4% (w/v) paraformaldehyde at room temperature for 20 min. After washing with PBS three times, the cells were permeabilized with 0.2% Triton X-100 for 20 min and then blocked with 10% FBS. The specific primary antibodies diluted in 10% FBS were added to the cells and incubated for 1 h at room temperature. After washing with PBS three times, the cells were incubated with appropriate secondary antibodies diluted in 10% FBS for 1 h at room temperature. The nuclei were stained with DAPI for 5 min at room temperature, mounted with Prolong Diamond (Invitrogen), and examined using a Zeiss LSM 800 confocal microscope. Filipin staining was performed as previously described [64].

## Cell surface biotinylation assay

Cells were washed twice with ice-cold PBS-CM (PBS plus 0.1 mM $CaCl_2$, 1 mM $MgCl_2$) and then incubated with 1 mg/mL EZ-Link NHS-biotin (Thermo) in PBS-CM at 4˚C for 30 min. The unreacted biotin was quenched with 10 mM glycine in PBS-CM at 4˚C for 10 min. Cells were then washed twice with ice-cold PBS-CM and incubated in prewarmed culture medium with 10 mM glycine at 37˚C for the indicated durations. Cytosolic and membrane fractions

were prepared as previously described [65]. Cytosol and membrane lysates were incubated with NeutrAvidin agarose (Thermo) and rotated at 4˚C for 2 h. The agarose beads were washed three times with homogenization buffer (10 mM HEPES pH 7.4, 10 mM KCl, 1.5 mM MgCl$_2$, 5 mM EDTA, 5 mM EGTA, and 250 mM sucrose). Biotinylated proteins were eluted with SDS-PAGE sample buffer and analyzed using immunoblotting.

### FRAP

Time series were carried out after HeLa cells were transiently transfected with AP2B1-m-Cherry for 24 h and then treated with P4 (10 μM) for another 8 h. Cells were imaged for 30 s, bleached for 30 s at a maximum laser intensity in a region of 7 μm × 7 μm, and imaged for an additional 5 min. Fluorescence recovery was analyzed by defining CCPs in the FRAP region before bleaching followed by measuring the fluorescence intensity in the FRAP region over time on a Zeiss LSM 800 confocal microscope.

### Calcium assay

PK-15 cells were loaded with membrane-permeable Fluo-3-AM (5 μM), a fluorescent Ca$^{2+}$ indicator, for 10 min at 37˚C in the dark. An excitation wavelength of 488 nm was provided by a Zeiss LSM 800 confocal microscope and fluorescence signals were examined using a 515-nm pass emission filter. Background fluorescence was subtracted from all signals. Changes in [Ca$^{2+}$] were represented using changes in fluorescence expressed as F/F0, where F0 is the control, diastolic fluorescence. The change in fluorescence intensity after drug treatments was normalized using the initial intensity.

### Lysosome distribution quantification

Lysosome distribution was analyzed in HeLa cells ranging 800–2,500 μm$^2$. The nuclear area was excluded during quantification. Average LAMP1 intensities were measured for the whole cell (I$_{total}$), area within 5 μm of the nucleus (I$_{perinuclear}$), and area >10 μm from the nucleus (I$_{peripheral}$). The perinuclear and peripheral normalized intensities were first calculated and normalized as I$_{<5}$ = I$_{perinuclear}$/I$_{total}$-100 and I$_{>10}$ = I$_{peripheral}$/I$_{total}$-100, respectively. The perinuclear index was defined as I$_{<5}$-I$_{>10}$ × 100. Quantifications were performed while blinded to the experimental groups presented.

### Lysosomal pH measurement

PK-15 cells were washed with HBSS buffer twice and incubated with PO2 (2 μM) for 30 min at 37˚C. After washing with HBSS buffer twice, the cells were incubated with buffers (135 mM KCl, 2 mM K$_2$HP$_4$O$_2$, 20 mM HEPES, 1.2 mM CaCl$_2$, and 0.8 mM MgSO$_4$) adjusted to a series of defined pH values (3, 3.5, 4, 4.5, 5, 5.5, 6, 7, and 7.5) for 5 min at room temperature and subjected to an excitation wavelength of 385 nm and emission wavelengths of 440 nm (F1) and 540 nm (F2) using a ZEISS LSM 800 confocal microscope. R = F2/F1 and pH standard curve was used to calculate lysosomal pH with ImageJ software.

### ELISA assay

GnrH, LH, FSH, P4 and E2 levels in murine serum were measured using ELISA kits (Bioswamp) according to the manufacturer instructions.

## Transmission electron microscopy

PK-15 cells were treated with P4 (10 μM) for 4 h followed by infection with PRV HN1201 (MOI = 100) at 4°C for 2 h in the continued presence of P4. The temperature was then adjusted to 37°C for 10 min to allow entry. After three washes with PBS, cells were fixed with 1% glutaraldehyde for 30 min, fixed with 1% osmium tetroxide for 1 h, washed with PBS, dehydrated, and embedded in EPON 812. Ultrathin sections were cut and stained with aqueous uranyl acetate and lead citrate. Images were taken on an FEI Tecnai G2 Spirit Transmission Electron Microscope operated at 120 kV.

## Histological analysis

Tissues dissected from mice were fixed in 4% paraformaldehyde (Sigma) overnight, embedded in paraffin, sectioned, and stained with hematoxylin (Sigma) and eosin (Sigma) solution.

## Statistical analysis

All data were analyzed using GraphPad Prism 8 software and a two-tailed Student's $t$-test. $P < 0.05$ was considered statistically significant. Data are shown as the mean ± standard deviation from three independent experiments. For mouse survival studies, Kaplan–Meier survival curves were generated and analyzed for statistical significance.

# Supporting information

**S1 Data. List of TRPML1-associated proteins identified by CoIP.**
(XLSX)

**S1 Table. List of primers used in this study.**
(DOCX)

**S2 Table. List of sgRNAs and shRNAs used in this study.**
(DOCX)

**S1 Fig. PRV infection downregulates the mRNA expression levels of GNRHR, LHCGR and FSHR.** (A) Female C57BL/6J mice were intranasally infected with PRV HN1201 ($1 \times 10^2$– $1 \times 10^4$ TCID$_{50}$ per mouse). Survival rate was monitored daily for 12 days (n = 12). (B) Female C57BL/6J mice were intranasally infected with PRV HN1201 ($2 \times 10^2$ per mouse) for 2 days. PRV gE expression in the ovary, uterus, hypothalamus, pituitary and trigeminal nerve was detected by immunohistochemistry. PRV-induced mouse histopathology was performed by hematoxylin and eosin staining. Scale bar: 50 μm. (C–E) The mRNA levels of GnRHR in the pituitary (C), and LHCGR (D) and FSHR (E) in the ovary of mock-infected or PRV-infected female mice were analyzed by qRT-PCR analysis at 2 days post pregnancy (n = 3). (F–I) LH (F), FSH (G) P4 (H) and E2 (I) in the serum of indicated female mice were quantified by ELISA assay at indicated pregnant days (n = 6). (J and K) The mRNA levels of LHCGR (J) and FSHR (K) in the ovary of indicated female mice were analyzed by qRT-PCR analysis at 4 days post pregnancy (n = 3). Data are expressed as the mean ± SD of 3 independent experiments. $P$-values were determined by Student's $t$-test. $P < 0.05$ was considered statistically significant. (TIF)

**S2 Fig. P4 injection is harmless to mice.** (A) Female C57BL/6J mice were injected daily with the indicated concentrations of P4. Body weight was measured from days 1 to 10 (n = 10). (B) F Female C57BL/6J mice were injected daily with P4 at doses of 0, 3, and 10 mg/kg per mouse. Hematoxylin and eosin staining was performed on sections from the myocardium, liver,

spleen, lung, kidney, and brain at 10 days post-injection. Scale bar: 50 μm. (C and D) On day -1, female C57BL/6J mice were either mock-infected or intranasally infected with PRV HN1201 ($2 \times 10^2$ TCID$_{50}$ per mouse). On day 0, the mock-infected or PRV-infected female mice were mated with male mice overnight. On day 2, the female mice were injected with leuprolide (1.5 mg/kg). On day 4, the mRNA levels of PTGS2 (C) and SCT (D) in the ovary were analyzed by qRT-PCR analysis. Data are expressed as the mean ± SD of 3 independent experiments. *P*-values were determined by Student's *t*-test. $P < 0.05$ was considered statistically significant.
(TIF)

**S3 Fig. P4 inhibits PRV proliferation which is not dependent on PGR.** (A) PK-15 cells were treated with the indicated concentrations of P4 (0–30 μM) for 12–72 hours. Cell viability was assessed by a CCK-8 assay. (B and C) PK-15 cells treated with vehicle, poly(I:C) (5 μg/mL), poly(dA:dT) (5 μg/mL), HT-DNA (2 μg/mL), or LPS (10 μg/mL) for 24 hours. The mRNA levels of IFN-β (B) and IL-6 (C) were analyzed by qRT-PCR. (D) Immunoblotting was performed to detect PGR in sgcontrol and sgPGR PK-15 cells. (E) sgcontrol and sgPGR PK-15 cells were infected with PRV HN1201 (MOI = 0.1) and treated with P4 (10 μM) for 24 hours. Viral titer was determined by a TCID$_{50}$ assay. (F) sgcontrol and sgPGR PK-15 cells were transfected with the PGR-Flag plasmid for 24 hours. The cells were then infected with PRV HN1201 (MOI = 0.1) and treated with P4 (10 μM) as indicated for 24 hours. PRV gE, PGR-Flag, and PGR were analyzed by immunoblotting. (G) PK-15 cells were incubated with PRV HN1201 (MOI = 0.1) and P4 (10 μM) at 4˚C for 2 hours. Viral attachment was detected by qRT-PCR analysis of viral genome copy numbers on the PM. (H) PK-15 cells were incubated with EdU-labeled PRV HN1201 (MOI = 0.1) and P4 (10 μM) at 4˚C for 2 hours. Viral attachment was observed using Apollo staining (red). DiI (green, 20 μM) indicated the PM. Scale bar: 10 μm. (I) Quantification of the relative EdU intensity on the PM of PK-15 cells from (H). Data are expressed as the mean ± SD of 3 independent experiments. *P*-values were determined by Student's *t*-test. $P < 0.05$ was considered statistically significant.
(TIF)

**S4 Fig. P4 induces the aberrant lysosomal function.** (A) PK-15 cells were treated with P4 (10 μM) for 8 hours. Co-localization analysis of intracellular cholesterol (filipin) with mitochondria, peroxisomes, endoplasmic reticulum (ER), and lysosomes was performed by immunofluorescence and filipin staining. Scale bar: 10 μm. (B) PK-15 cells were treated with vehicle or P4 (10 μM). Lysosomal pH was measured 8 hours post-treatment. (C) PK-15 cells were transfected with a vector or NPC1-GFP plasmid for 24 hours. Afterward, cells were treated with vehicle or P4 (10 μM) for an additional 8 hours. Co-localization analysis of LAMP1 (red) with NPC1-GFP was performed by immunofluorescence analysis. Scale bar: 10 μm. (D) Quantification of the perinuclear index of LAMP1 in PK-15 cells from (C). (E) PK-15 cells were treated with P4 (10 μM) for 0–8 hours. Co-localization analysis of fluo-3 with LAMP1 was performed by immunofluorescence analysis. Scale bar: 10 μm. (F) Pearson's correlation coefficients of fluo-3 with LAMP1 in PK-15 cells from (E). Data are expressed as the mean ± SD of 3 independent experiments. *P*-values were determined by Student's *t*-test. $P < 0.05$ was considered statistically significant.
(TIF)

**S5 Fig. P4-induced aberrant lysosomal function is rescued by TRPML1.** (A and B) PK-15 cells were infected with PRV HN1201 (MOI = 0.1) for 0–24 hours. The mRNA levels of TPCN1 (A) and TPCN2 (B) were analyzed by qRT-PCR. (C) PK-15 cells were treated as indicated in (A). TPCN1 and TPCN2 proteins were analyzed by immunoblotting. (D) Mice were

either mock-infected or intranasally infected with PRV ($2 \times 10^3$ TCID$_{50}$ per mouse) for 2 days. TPCN1 and TPCN2 in the uterus were analyzed by immunoblotting (n = 3). (E) PK-15 cells were transfected with TRPML1-FLAG, TRPML1-FLAG L106P, TPCN1-FLAG, and TPCN2-FLAG plasmids and treated with vehicle or P4 (10 μM) for 24 hours. Colocalization of LAMP1 with TRPML1 variants, TPCN1, and TPCN2 was analyzed by immunofluorescence analysis. Scale bar: 10 μm. (F) Quantification of the perinuclear index of LAMP1 in PK-15 cells from (E). (G) PK-15 cells were transfected with TRPML1-mCherry, TRPML1-mCherry L106P, TPCN1-mCherry, and TPCN2-mCherry and treated with vehicle or P4 (10 μM) for 24 hours. Colocalization of fluo-3 and intracellular cholesterol (filipin) with TRPML1 variants, TPCN1, and TPCN2 was analyzed by immunofluorescence. Scale bar: 10 μm. (H) TRPML1 in scramble, shTRPML1-1, and shTRPML1-2 was detected by immunoblotting. (I) PK-15 cells were treated with P4 (10 μM), ML-SA1 (20 μM), and ML-SI1 (10 μM) as indicated. Fluo-3 and filipin staining were performed 8 hours post-treatment. Scale bar: 10 μm. (J) PK-15 cells were incubated with PRV HN1201 (MOI = 0.1) and treated with vehicle, P4 (10 μM), ML-SA1 (20 μM), or ML-SI1 (10 μM) as indicated at 4°C for 2 hours. Cells were then shifted to 37°C for 10 minutes to allow entry. After washing with trypsin (1 mg/mL) to remove residual virions on the PM, viral entry was detected by qRT-PCR analysis of viral genome copy numbers in the cells. (K) PK-15 cells were treated with vehicle, P4 (10 μM), ML-SA1 (20 μM), or ML-SI1 (10 μM) as indicated for 8 hours and then infected with PRV HN1201 (MOI = 0.1) for 24 hours. p-DNM1, DNM1, and PRV gE were detected by immunoblotting. (L) TRPML1 in *Mcoln1*$^{+/+}$ and *Mcoln1*$^{-/-}$ MEF cells was detected by immunoblotting. Data are expressed as the mean ± SD of 3 independent experiments. *P*-values were determined by Student's *t*-test. *P* < 0.05 was considered statistically significant.
(TIF)

**S6 Fig. PRV infection promotes PI(3,5)P$_2$ synthesis.** (A) Schematic diagram of the PI(3,5)P$_2$ metabolic pathways. (B–D) PK-15 cells were infected with PRV HN1201 (MOI = 0.1) for the indicated times. The mRNA levels of PIKfyve (B), MTM1 (C), and Fig 4 (D) were analyzed by qRT-PCR analysis. (E) Immunoblotting analysis of NPC1 (lysosome), Calnexin (ER), Tom20 (mitochondria), and Lamin B1 (nucleus) in PK-15 cell lysates subjected to iodixanol density gradient centrifugation. Data are expressed as the mean ± SD of 3 independent experiments. *P*-values were determined by Student's *t*-test. *P* < 0.05 was considered statistically significant.
(TIF)

**S7 Fig. P4 stimulates the ubiquitination and proteasomal degradation of TRPML1 through MDM2.** (A) PK-15 cells were treated with P4 (10 μM) for 0–24 hours. The mRNA levels of TRPML1 were analyzed by qRT-PCR. (B) PK-15 cells were treated with P4 (0–30 μM) for 24 hours. TRPML1 levels were analyzed by immunoblotting. (C) PK-15 cells were treated with P4 (10 μM) for 0–24 hours. TRPML1 levels were analyzed by immunoblotting. (D) Mice were injected daily with vehicle or P4 (10 mg/kg per mouse) for 10 days. TRPML1 levels in the uterus were analyzed by immunoblotting (n = 3). (E) PK-15 cells were treated with vehicle, P4 (10 μM), MG-132 (10 μM), or chloroquine (10 μM) as indicated for 0–24 hours. TRPML1 levels were analyzed by immunoblotting. (F) PK-15 cells were transfected with TRPML1-FLAG and HA-UB plasmids, and treated with P4 (0–30 μM) as indicated for 24 hours. Ubiquitination of TRPML1-FLAG was analyzed by ubiquitination assay. (G) PK-15 cells were transfected with TRPML1-FLAG variants and HA-UB plasmids, and treated with P4 (0–30 μM) for 24 hours. Ubiquitination of TRPML1-FLAG was analyzed by ubiquitination assay. (H) Coomassie blue staining was used to detect potential E3 ubiquitin ligases associated with TRPML1. (I) PK-15 cells were transfected with TRPML1-FLAG and the indicated E3 plasmids for 24 hours. The interactions of TRPML1-FLAG with the indicated E3 ligases were analyzed by CoIP

analysis. (J) PK-15 cells were treated with P4 (10 μM) for the indicated times. MDM2 and TRPML1 levels were analyzed by immunoblotting. (K) MDM2 and TRPML1 levels in the uterus from experiment D were analyzed by immunoblotting. Data are expressed as the mean ± SD of 3 independent experiments. $P$-values were determined by Student's $t$-test. $P < 0.05$ was considered statistically significant.
(TIF)

## Acknowledgments

We thank Dr. You-Bao Zhao for valuable advice and discussions.

## Author Contributions

**Conceptualization:** Jiang Wang, Sheng-Li Ming, Bei-Bei Chu.

**Data curation:** Bing-Qian Su, Guo-Yu Yang, Sheng-Li Ming.

**Formal analysis:** Bing-Qian Su, Guo-Yu Yang, Sheng-Li Ming.

**Funding acquisition:** Bei-Bei Chu.

**Investigation:** Bing-Qian Su.

**Methodology:** Bing-Qian Su.

**Project administration:** Guo-Yu Yang, Jiang Wang, Sheng-Li Ming, Bei-Bei Chu.

**Supervision:** Jiang Wang, Sheng-Li Ming, Bei-Bei Chu.

**Validation:** Bing-Qian Su, Guo-Yu Yang.

**Visualization:** Bing-Qian Su.

**Writing – original draft:** Jiang Wang, Bei-Bei Chu.

**Writing – review & editing:** Jiang Wang, Bei-Bei Chu.

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
