## [Decision Letter · Decision Letter 0]

21 Nov 2023

Dear Dr. Chu,

Thank you very much for submitting your manuscript "Virus inhibits progesterone-induced inactivation of TRPML1 to facilitate viral entry" for consideration at PLOS Pathogens. As with all papers reviewed by the journal, your manuscript was reviewed by members of the editorial board and by several independent reviewers. In light of the reviews (below this email), we would like to invite the resubmission of a significantly-revised version that takes into account the reviewers' comments.

Dr. Chu,

Thank you for resubmitting the manuscript with additional details. We have sent the manuscript to three experts in the field that found the work compelling. While some of the reviewers comments were varied in opinion, the general consensus is that the work needs additional clarification on distinct aspects of the experiments. As such, the editorial team has come to a decision of Major Modify for this manuscript. If you feel that the concerns raised can be addressed we would look forward to reexamining this manuscript for PLoS Pathogens. Thank you for your patience in this matter.

Cheers,

Eain Murphy Ph.D.

We cannot make any decision about publication until we have seen the revised manuscript and your response to the reviewers' comments. Your revised manuscript is also likely to be sent to reviewers for further evaluation.

Sincerely,

Eain A Murphy, Ph.D.

Academic Editor

PLOS Pathogens

Blossom Damania

Section Editor

PLOS Pathogens

Kasturi Haldar

Editor-in-Chief

PLOS Pathogens

orcid.org/0000-0001-5065-158X

Michael Malim

Editor-in-Chief

PLOS Pathogens

orcid.org/0000-0002-7699-2064

Dr. Chu,

Thank you for resubmitting the manuscript with additional details. We have sent the manuscript to three experts in the field that found the work compelling. While some of the reviewers comments were varied in opinion, the general consensus is that the work needs additional clarification on distinct aspects of the experiments. As such, the editorial team has come to a decision of Major Modify for this manuscript. If you feel that the concerns raised can be addressed we would look forward to reexamining this manuscript for PLoS Pathogens. Thank you for your patience in this matter.

Cheers,

Eain Murphy Ph.D.

Reviewer's Responses to Questions

**Part I - Summary**

Reviewer #1: In this manuscript, the authors used pseudorabies virus (PRV), a swine alphaherpesvirus, and mice model to address why PRV infection causes reproductive failure (infertility). Their data showed that PRV infection affected the hypothalamus-pituitary-ovary axis (HPOA) and induced infertility by downregulating progesterone P4, and further revealed the molecular pathway contributing this phenomenon, proposing a novel mechanism by which PRV hijacks the lysosomal pathway to evade P4-mediated antiviral defense and impair female fertility. This study is of general significance for explaining the impact of viral diseases on female reproductive health and for developing antiviral therapies to control the associated reproduction issues clinically caused by virus infection. The work is well designed and complete, and the data supporting the conclusion are basically sufficient.

Reviewer #2: In the submitted manuscript, Su et al. investigates the important regulatory role of P4 in the process of infertility induced by PRV virus. Importantly, endo-lysosomal channel TRPML1 as a host factor to facilitate viral entry was demonstrated, and P4 can induce proteasomal degradation of TRPML1 via MDM2 to inhibit viral entry, which reveals a novel mechanism of how PRV influenced P4 to induce infertility and provides a new strategy for developing antiviral therapies. Overall, the study contributes a new knowledge on the role of P4 and TRPML1 in viral endosomal entry pathways. The conclusions are backed up by a lot of data. However, there are some points that would need addressing before publication.

Reviewer #3: Strengths

1. Identification that PRV facilitates lysosomal-dependent viral entry via TRPML1

2. Exploration of the effects of leuprolide in this infection model is interesting

Weaknesses

1. Insufficient details regarding the mouse model of PRV infection

2. Missed opportunity to explore the effects of PRV infection in mouse CNS and reproductive organs

3. Without better delineation of PRV-induced mouse histopathology, initial studies are largely observational and the more mechanistic studies (i.e., effects of PRV infection on TRPML1 expression and lysosomal synthesis of PI synthesis) lack sufficient context to determine relevancy

4. Insufficient explanation for the rising FSH levels measured in mock-infected controls during the first 12 days of pregnancy

5. While manuscript contains few grammatical errors it is sometimes difficult to follow and more importantly lacks adequate explanation of the significance of its findings

6. Figure panel size makes data interpretation challenging and legends do not provide sufficient detail (including but not limited to statistical considerations)

**Part II – Major Issues: Key Experiments Required for Acceptance**

Reviewer #1: No

Reviewer #2: 1. In Fig. 1, it was shown that 2 of 12 were pregnant in the PRV-infected group, why the Embryo number of PRV-infected mice was 3? Did the authors put the embryos of different batches of pregnant and virus-infected mice together?

2. In Fig. 5, the authors have shown that PRV infection upregulated TRPML1 expression and the TRPML1 protein plays an important role in viral entry. Previous studies have found that endosomal cation channel TRPML2 (mBio. 2018 Jan 30; 9(1): e02314-17. doi: 10.1128/mBio.02314-17), as a key host factor, can also promote viral entry by enhancing the efficiency of viral trafficking. Why did the authors not consider the response of other TRPML proteins to PSV?

3. In Fig. 4 and Fig. 5, it was shown that PRV infection stimulated significant Ca2+ increases, and this enhancement was mainly through TRPML1 channel regulated by PRV infection. The effect of TRPML1 channel activity on viral infection should be investigated.

4. PRV infection not only reduces the level of P4, but also promotes the expression of TRPML1 and intracellular calcium transport. How to distinguish and verify which process happens first on a time scale? At the same time, in addition to affecting TRPML1 expression, whether P4 can also affect TRPML1-mediated intracellular calcium and cholesterol accumulation?

Reviewer #3: Figure 1 provides underpinning for studies - that PRV infection disrupts HPO axis function. As displayed in panel G, mock-infected controls show serum levels of FSH that rise in linear fashion from day 0 - day 12 of pregnancy. This result to me is unexpected. It would be expected for FSH levels to be highest on day 0 and at parturition (not measured in this study) and also that rising concentrations of estrogen and progesterone during pregnancy would inhibit FSH release and lower FSH levels from their day 0 peak. Manuscript does not address FSH results in the mock-infected controls or possible reasons why mice in their study demonstrate steadily rising levels of FSH during pregnancy.

In mice, PRV causes CNS symptoms and high mortality. After oronasal inoculation, PRV replicates in nasal epithelium and enters free nerve endings in TG and other neurons innervating nasal mucosa. PRV replication in the cell bodies of TG is followed by axonal transport to second-order neurons in the afferent trigeminal nuclei of the brain stem. Experimental detail regarding lethality should be included (especially as mice were infected with PRV HN1201, a strain that induces more severe infection in swine compared to the classical Fa strain).

Likewise, it will be important to assess histopathology of PRV infection in CNS tissue (including the hypothalamus and pituitary) and reproductive tissue (PRV causes viremia that can seed endometrial tissue and ovaries). These studies are need to clarify the effects of PRV infection on the hypothalamus or pituitary vs. ovary vs. PRV-induced fetal demise (all of which have the potential to reduce circulating levels of progesterone.

**Part III – Minor Issues: Editorial and Data Presentation Modifications**

Reviewer #1: 1. Only a alphaherpesvirus (PRV) was used in this study. The title should define “Pseudorabies virus”.

2. In the results, a summary for the data and a diagram describing the mechanism should be provided as the last paragraph of this section in order to help readers understand the findings.

3. There are some inappropriate descriptions and grammar uses in the manuscript.

Reviewer #2: 1. In mouse model, why did the author set 12 days as the detection time of mouse pregnancy?

2. In Fig. 1 and 2, the author has introduced the downstream of hypothalamus-pituitary-ovary axis includes estrogen and P4, why only P4 level was tested and verified, and did PSV infection cause changes in estrogen level?

3. In Fig. 5C, why is there no detectable protein from the virus in the first six hours?

4. Abbreviations should be introduced upon first mention.

5. In Fig. 1, why does P4 level decrease less significantly to GnRH, LH, FSH in hypothalamus-pituitary-ovary axis?

6. The schematic model showing PRV evades P4-mediated antiviral defense and impairs female fertility in Fig. S8 should be moved to the main manuscript.

7. P12 line 250 - rephrase this sentence - " However, MCOLN1 was transcriptionally activated by PRV infection (Figure 5A and 5B). TRPML1 protein level was also increased in PRV-infected cells or in a murine uterus".

8. The authors have shown that PRV infection specifically resulted in translocation of lysosomal cholesterol and Ca2+ to the PM, how does increased calcium concentration promote the viral entry.

9. I do think a professional copy edit is still required, and it would make the manuscript much smoother to read.

Reviewer #3: Data is poorly displayed throughout manuscript and legend content lacks sufficient detail throughout. These are major rather than minor concerns as it makes manuscript difficult to evaluate. As an additional major concern, sentence construction is sometime disjointed and various statements in the introduction and discussion lack cogency and precision. As example, lines 381-384:

"Pregnant women are the largest group of adults with disseminated HSV infection. The overall HSV seroprevalence among pregnant women is 72%. The mean gestational age at presentation is 31 weeks and the case fatality rate is 39% for both mothers and neonates."

These 3 sentences seem to have little to do with each other and with the research presented in the manuscript. If they are included in the discussion, tt would also be useful to provide reference for the statements that pregnant women are the largest group of adults with disseminated HSV infection and that rates of fatality are identical among HSV-infected women and neonates.

PLOS authors have the option to publish the peer review history of their article (what does this mean?). If published, this will include your full peer review and any attached files.

Reviewer #1: No

Reviewer #2: No

Reviewer #3: No
---

## [Decision Letter · Decision Letter 1]

8 Jan 2024

Dear Dr. Chu,

We are pleased to inform you that your manuscript 'Pseudorabies virus inhibits progesterone-induced inactivation of TRPML1 to facilitate viral entry' has been provisionally accepted for publication in PLOS Pathogens.

Best regards,

Eain A Murphy, Ph.D.

Academic Editor

PLOS Pathogens

Blossom Damania

Section Editor

PLOS Pathogens

Kasturi Haldar

Editor-in-Chief

PLOS Pathogens

orcid.org/0000-0001-5065-158X

Michael Malim

Editor-in-Chief

PLOS Pathogens

orcid.org/0000-0002-7699-2064

Dr Chu,

Thank you for your re-submission and your concerted efforts to accommodate the concerns raised in the initial submission. The editors have come to a decision of Accept for this manuscript. Congratulations.

Chjeers,

Eain Murphy

Reviewer Comments (if any, and for reference):

Reviewer's Responses to Questions

**Part I - Summary**

Reviewer #1: This submission is of general significance for explaining the impact of viral diseases on female reproductive health and for developing antiviral therapies to control the associated reproduction issues clinically caused by virus infection. The novelty is to identify a novel mechanism by which PRV hijacks the lysosomal pathway to evade P4-mediated antiviral defense and impair female fertility.

Reviewer #2: The authors have well responded to my all comments and revised the related parts. I have no any additional comments or concerns.

**Part II – Major Issues: Key Experiments Required for Acceptance**

Reviewer #1: The revised version has added corresponding data to meet the reviewers' requirements.

Reviewer #2: No

**Part III – Minor Issues: Editorial and Data Presentation Modifications**

Reviewer #1: The revised version has presented corresonding modifications to increase the statements of the text.

Reviewer #2: No

PLOS authors have the option to publish the peer review history of their article (what does this mean?). If published, this will include your full peer review and any attached files.

Reviewer #1: No

Reviewer #2: No

---

## [Editor Report · Acceptance letter]

17 Jan 2024

Dear Dr. Chu,

We are delighted to inform you that your manuscript, " Pseudorabies virus inhibits progesterone-induced inactivation of TRPML1 to facilitate viral entry ," has been formally accepted for publication in PLOS Pathogens.

Best regards,

Michael Malim

Editor-in-Chief

PLOS Pathogens

orcid.org/0000-0002-7699-2064